# Effect of Environmental Conditions on the Yield of Peel and Composition of Essential Oils from Citrus Cultivated in Bahia (Brazil) and Corsica (France)

**François Luro [1,*], Claudia Garcia Neves [2], Gilles Costantino [1], Abelmon da Silva Gesteira [3], Mathieu Paoli [4], Patrick Ollitrault [5], Félix Tomi [4], Fabienne Micheli [2,6] and Marc Gibernau [4]**

[1] Unité Mixte de Recherche Amélioration Génétique et et Adaptation des Plantes (UMR AGAP) Corse, Institut National de Recherche pour l'Agriculture, l'Alimentation et l'Environnement (INRAE), 20230 San Giuliano, France; gilles.costantino@inrae.fr

[2] Centro de Biotecnologia e Genética (CBG), Departamento de Ciências Biológicas (DCB), Universidade Estadual de Santa Cruz (UESC), Rodovia Ilhéus-Itabuna, km 16, Ilhéus, BA 45662-900, Brasil; claudia-garcia23@hotmail.com (C.G.N.); fabienne.micheli@cirad.fr (F.M.)

[3] Empresa Brasileira de Pesquisa e Agropecuária (EMBRAPA) Mandioca e Fruticultura, Rua Embrapa, s/nº, Cruz das Almas, BA 44380-000, Brasil; abelmon.gesteira@embrapa.br

[4] Equipe Chimie et Biomasse, Unité Mixte de Recherche 6134 SPE, Université de Corse-CNRS, Route des Sanguinaires, 20000 Ajaccio, France; paoli_m@univ-corse.fr (M.P.); tomi_f@univ-corse.fr (F.T.); gibernau_m@univ-corse.fr (M.G.)

[5] Unité Mixte de Recherche Amélioration Génétique et et Adaptation des Plantes (UMR AGAP) Corse, Centre de coopération Internationale en Recherche Agronomique pour le développement (CIRAD), 20230 San Giuliano, France; patrick.ollitrault@cirad.fr

[6] Unité Mixte de Recherche Amélioration Génétique et et Adaptation des Plantes (UMR AGAP), Montpellier, Centre de coopération Internationale en Recherche Agronomique pour le développement (CIRAD), 34398 Montpellier, France

\* Correspondence: francois.luro@inrae.fr; Tel.: +33-4-95-59-59-46

**Abstract:** The cosmetic and fragrance industry largely exploits citrus essential oils (EOs) because of their aromatic properties. EO compositions are complex and differ between fruit pericarp (PEO) and leaf (LEO). Citrus fruit grow in many countries under very different climates. Seventeen citrus cultivars were selected and their similarities between the two collections were verified by SSR (Single Sequence Repeat) and InDel (Insertion and Deletion) markers to assess the effects of the environment and cultivation practices on the EO yield and composition. LEOs and PEOs were extracted by water distillation and analyzed by GC-MS. PEO yields were generally higher in Corsica than in Bahia, especially in the citron family. PEOs in this family were richer in limonene in Bahia than in Corsica while, conversely, neral, geranial and derivatives were present in a higher proportion in Corsican varieties. A few minor components were site-specific, such as nookaton, a pummelo-specific compound that was not present in grapefruit cultivated in Bahia. If climate change over the last 20 years has not affected the PEO composition in Corsica, the contrasted environmental conditions and cultural practices between Bahia and Corsica could possibly explain the EO variations.

**Keywords:** aromatic compounds; CPG-SM; climate; cultural practices; genetic diversity

## 1. Introduction

Citrus fruit especially grow in different humid tropical, dry tropical and Mediterranean climatic conditions. They are mostly poorly adapted to freezing temperatures but can still withstand climates with relatively cold winters (average temperatures between 0 and 10 °C), e.g., in the Mediterranean area and Northeast Asia (e.g., Korea, Japan). Some species such as *C. reticulata* (mandarins) or *Fortunella* sp. (Kumquats) originating from China are considered to be better adapted to climates with cool winter temperatures, which favor the red orange fruit coloring and increased fruit pulp acidity [1–4]. Pummelos (*C. maxima*) originating from tropical areas (e.g., Thailand, Indonesia, Malaysia), generally have higher heat requirements than other citrus fruit for ripening and meeting the taste expectations of the usual consumers [5]. "Key" lime (*C. aurantifolia*) and "Bears" lime (*C. latifolia*) are also mainly grown in warm regions (Mexico, Caribbean area, Brazil, Florida, southern Arabian Peninsula). The location where limes are cultivated affects the green external fruit color needed for marketing—cold temperatures that trigger the chlorophyll degradation are harmful for this crop [5]. Conversely, lemon (*C. limon*) is mainly grown in cooler regions (Argentina, Italy, Spain, Turkey) where the relatively low temperatures increase the acidity and yellow fruit color intensity [6]. Significant variations in secondary metabolite contents, such as carotenoids, were also reported between Mediterranean and tropical areas [7]. Mediterranean conditions amplified interspecific differentiation, especially by increasing the beta-cryptoxanthin and cis-violaxanthin contents in oranges (*C. sinensis*) and beta-carotene and phytoene-phytofluene contents in mandarins. The environment therefore modulates the expression of several citrus characters, but how does it affect the aroma and essential oil (EO) composition? EOs are present in the leaves, flowers and the outer part of the skin of the fruit (flavedo) and are characteristic of citrus fruit and the Rutaceae family [8]. Citrus EOs are extensively exploited by the cosmetic industry and by the food flavoring sector [9]. Although fruit EO is mostly used by industry, leaf EO, called "petit grain" is also used in cosmetics.

Many factors are responsible for EO composition variations, some are endogenous (e.g., genetic variability, plant organ, fruit maturity stage and rootstock), while others are exogenous (e.g., diseases, cultural practices, climate and soil properties) [10]. In *Citrus*, ancestral species evolved in separate geographical areas during an allopatric phase prior to colonization of the same places where interspecific hybrids were generated [11]. The EO composition is consequently highly specific to each species, with specific compounds or proportions of common compounds [12,13]. Therefore, EOs are often used to assess the genetic diversity of species, quantify relationships between cultivars or species, and classify unknown cultivars on the basis of discriminating compounds [12,14–23]. These studies showed that leaf-extracted EO (LEO) is better suited to study diversity and taxonomy than fruit skin EO (PEO) because limonene is present in much lower proportions in LEO than in PEO, making it easier to observe variation of other compounds.

Fruit ripening is another factor responsible for changes in the EO composition [24–27]. The EO yield and composition during fruit ripening fluctuates according to the citrus cultivar [28]. EOs obtained from Pompia fruit growing in Sardinia (Italy) harvested in February presented higher concentrations of limonene, α-pinene, myrcene and (Z)-β-ocimene, than in other months [29]. A study of four Tunisian citrus cultivars found that monoterpene hydrocarbon levels were higher at the mature stage, while oxygenated monoterpenes were higher at the immature stage [30]. In the same study, EO yields were higher at different stages of fruit maturity according to the citrus cultivar: immature stage for lemon, mature stage for sour orange (*C. aurantium*) and semi-mature stage for mandarin and orange [30].

The rootstock can also influence the EO composition and yield. A study of EOs from bergamote (*C. bergamia*) grafted on four different rootstocks revealed similar linalool and linalyl acetate contents for Alemow (*C. macrophylla*) and Volkamer lemon (*C. limonia*) trees grafted on sour orange, while the amount of these two compounds and their oxygenated forms were reduced on trifoliate orange rootstock (*Poncirus trifoliata*) [31]. Compared to sour orange and "Swingle" citrumelo (*C. paradisi* × *P. trifoliata*), "Troyer" citrange (*C. sinensis* × *P. trifoliata*) rootstock increased monoterpene and linalool levels of kumquat fruit peel EOs (PEOs) [32], while "Swingle" citrumelo rootstock increased the

aldehyde levels of "Page" mandarin PEOs [33]. Benjamin et al. [34] concluded that the effects of rootstocks on citrus fruit flavor depended on specific rootstock/scion interactions.

Many exogenous factors are involved in EO composition variations. Diseases often alter the physiology and metabolism of host plants and often cause morphological and chemical changes in the different fruit tissues. For example, Huanglongbing (HLB), one of the most severe citrus diseases caused by the phloem bacterium *Candidatus* Liberibacter spp., affects the juice quality and PEOs, which contain few ethyl esters and lot of monoterpenes [35]. In other studies, the proportion of PEOs extracted from infected fruit was 30% lower than from healthy fruit [36], and specific compounds in fruit affected by HLB (two terpenes and one aldehyde) have been identified [37].

Deterre et al. [38], Dugo et al. [9,39], and Kirbaslar and Kirsbaslar [40], obtained clear evidence that geographical origin is one of the sources of variation in the EO composition. Sour orange marker compound quantities were different in EOs from three different geographical zones (Florida, Equator and Mediterranean), suggesting that the chemical profile could be a suitable marker of the geographical origin of EOs [38]. Variations in EO composition due to climatic conditions (temperature, day length, light, and water level), cultivation conditions (plant density, soil properties, soil type, and soil fertility), and cultivation practices (irrigation dose, fertilization, and mineral nutrition) are fully documented for medicinal and aromatic plants (for review [41]) but much less so for citrus.

It is not always easy to identify real variation factors by comparing different studies because the genetic conformity of studied citrus cultivars is not often verified and there is little information on the cultivation and environmental (climate and soil) conditions. We therefore aimed to study the interaction between genetic background and environmental conditions by selecting several cultivars from major citrus crops maintained in two collections from Brazil (EMBRAPA research station, Cruz das Almas, Bahia) and France (INRAE-CIRAD citrus collection, San Giuliano, Corsica) to study their leaf and fruit EO compositions. We verified the identities of the citrus batches by genotyping with SSR (Single Sequence Repeat) and InDel (Insertion and Deletion) markers and described the cultivation practices and variations in climatic conditions (tropical and Mediterranean) throughout the fruit development time.

## 2. Material and Methods

### 2.1. Plant Material

Seventeen cultivars of the *Citrus* genus representing 8 citrus species growing in two germplasm repositories, one in Brazil and one in France (Table 1) were selected based on their diversity. The batch of 17 cultivars was supplemented by 5 additional cultivars representing 3 *Citrus* species in order to have a better assessment of the genetic diversity of the sample selected for the study of EOs, and its representation in the overall diversity of the *Citrus* genus. Then 3 pummelo cultivars (*C. maxima*) ("Reinking", ICVN0100323, "Siamese", ICVN0101126 and "Tahiti", ICVN0100727) and *C. micrantha* (ICVN0101115) were added in order to have references of all the *Citrus* ancestral species. The Mexican lime (*C. aurantifolia*) accession (ICVN0100140) completed the additional samples, because it is one of the "Tahiti" lime parents [42,43]. These 5 additional citrus cultivars from INTRAE-CIRAD germplasm were not used for the EO composition analysis because they are not present in the EMBRAPA collection at Cruz das Almas. In France, the citrus repository was the Citrus Biological Resource Centre of pathogen-free citrus germplasm (BRC CITRUS, INRAE-CIRAD, NFS96-900) based in San Giuliano (Corsica) (latitude 42°17′ N, longitude 9°32′ E) [44]. In Brazil, the citrus germplasm repository was based at EMBRAPA *Mandioca e Fruticultura* located in Cruz das Almas, Bahia (12°40′39″ S latitude, 39°06′23″ W longitude and 225 m elevation) [45].

**Table 1.** List of citrus cultivars used in the different study analyses.

| Group | Cultivar | Taxonomy (Tanaka) | Corsican Id. | Brazilian Id. | Rootstock [a] | | Analyses | | |
|---|---|---|---|---|---|---|---|---|---|
| | | | | | Corsica | Bahia | Div [b] | LEO | PEO |
| Sour orange | Granito | *C. aurantium* L. | ICVN0110015 | Bag(2)12-71 | CC | RL | x | x | x |
| Citron | Etrog | *C. medica* L. | ICVN0100709 | Bag(2)31-183 | VL | RL | x | x | x |
| Lemon | Feminello | *C. limon* (L.) Burm. | ICVN0100180 | Bag(2)32-192 | SO | RL | x | x | x |
| Lemon | Eureka | *C. limon* (L.) Burm. | ICVN0100004 | Bag(2) | SO | RL | x | x | x |
| Lime | Tahiti | *C. latifolia* Tan. | ICVN0100058 | Bag(5) | SO | RL | x | x | x |
| Clementine | Tomatera | *C. clementina* Hort. ex Tan. | ICVN0100535 | Bag(2)23-133 | TO | RL | x | x | x |
| Mandarin | Dancy | *C. tangerina* Hort. ex Tan. | ICVN0100051 | Bag(1)15-586 | TO | RL | x | x | x |
| Mandarin | Fairchild | *C. reticulata* Blanco | ICVN0100030 | Bag(2)18-104 | TO | RL | x | | x |
| Mandarin | Willowleaf | *C. deliciosa* Ten. | ICVN0100133 | Bag(2)17-667 | TO | RL | x | x | x |
| Mandarin | Murcott | *C. reticulata* Blanco | ICVN0100601 | Bag(1)16-634 | CC | RL | x | x | |
| Mandarin | Sunki | *C. sunki* Hort. ex Tan. | ICVN0100705 | Bag(1)452 | TO | RL | x | x | x |
| Mandarin | Page | *C. reticulata* Blanco | ICVN0100159 | Bag(1)15-585 | TO | RL | x | x | x |
| Mandarin | Hybrida | *C. reticulata* Blanco | ICVN0100714 | | TO | RL | x | | x |
| Mandarin | Nasnaran | *C. amblycarpa* (Hassk.) Ochse | ICVN0100896 | Bag(4)2-9 | TO | RL | x | x | |
| Orange | Hamlin | *C. sinensis* (L.) Osb. | ICVN0100041 | Bag(2)9-52 | TO | RL | | | x |
| Orange | Valencia late | *C. sinensis* (L.) Osb. | ICVN0100246 | Bag(2)10-57 | TO | RL | x | x | x |
| Grapefruit | Pink ruby | *C. paradisi* Macf | ICVN0100605 | Bag(2)25-150 | CC | RL | x | | x |
| Pummelo | Reinking | *C. maxima* (Burm.) Merr. | ICVN0100323 | | TO | | x | | |
| Pummelo | Siamese | *C. maxima* (Burm.) Merr. | ICVN0101126 | | TO | | x | | |
| Pummelo | Tahiti | *C. maxima* (Burm.) Merr. | ICVN0100727 | | TO | | x | | |
| Micrantha | | *C. micrantha* Wester | ICVN0101115 | | VL | | x | | |
| Lime | Mexican | *C. aurantifolia* (Christm.) Swing. | ICVN0100140 | | CC | | x | | |

[a] CC: "Carrizo" citrange, VL: "Volkamer" lemon, SO: Sour orange, TO: Trifoliate orange, RL: "Rangpur" lime; [b] Div: genetic diversity analysis; LEO: leaf essential oil; PEO: peel essential oil; x: performed analysis.

## 2.2. Climate, Soil and Cultivation Practice Description

The average annual temperature at the INRAE-CIRAD research station was 15.2 °C. Monthly mean temperatures from the 2015 blossom period to the next year ranged from 31 °C to 4 °C (Figure 1). The average annual temperature at the EMBRAPA research station (Cruz das Almas-CDA) was about 24 °C. Monthly mean temperatures at the EMBRAPA research station from blossom period to the next year ranged from 26.5 °C to 21.8 °C (Figure 1).

The average annual rainfall was 800 mm (658 mm during the study period, i.e., April 2015 to March 2016) but the summer fruit growth period was dry in Corsica (Figure 1), with the rains occurring mainly at the end of autumn and winter. However, the trees were irrigated by microsprinklers during the dry period, according the estimated needs of the tree with respect to the potential evapotranspiration (Etp), rainfall (P) and cultural coefficient (Kc): (Etp–P) x Kc [46]. At the EMBRAPA research station, the average annual rainfall was about 1100 mm, with the rainfall decreasing during the fruit growth period (Figure 1). Trees were not irrigated at the EMBRAPA research station.

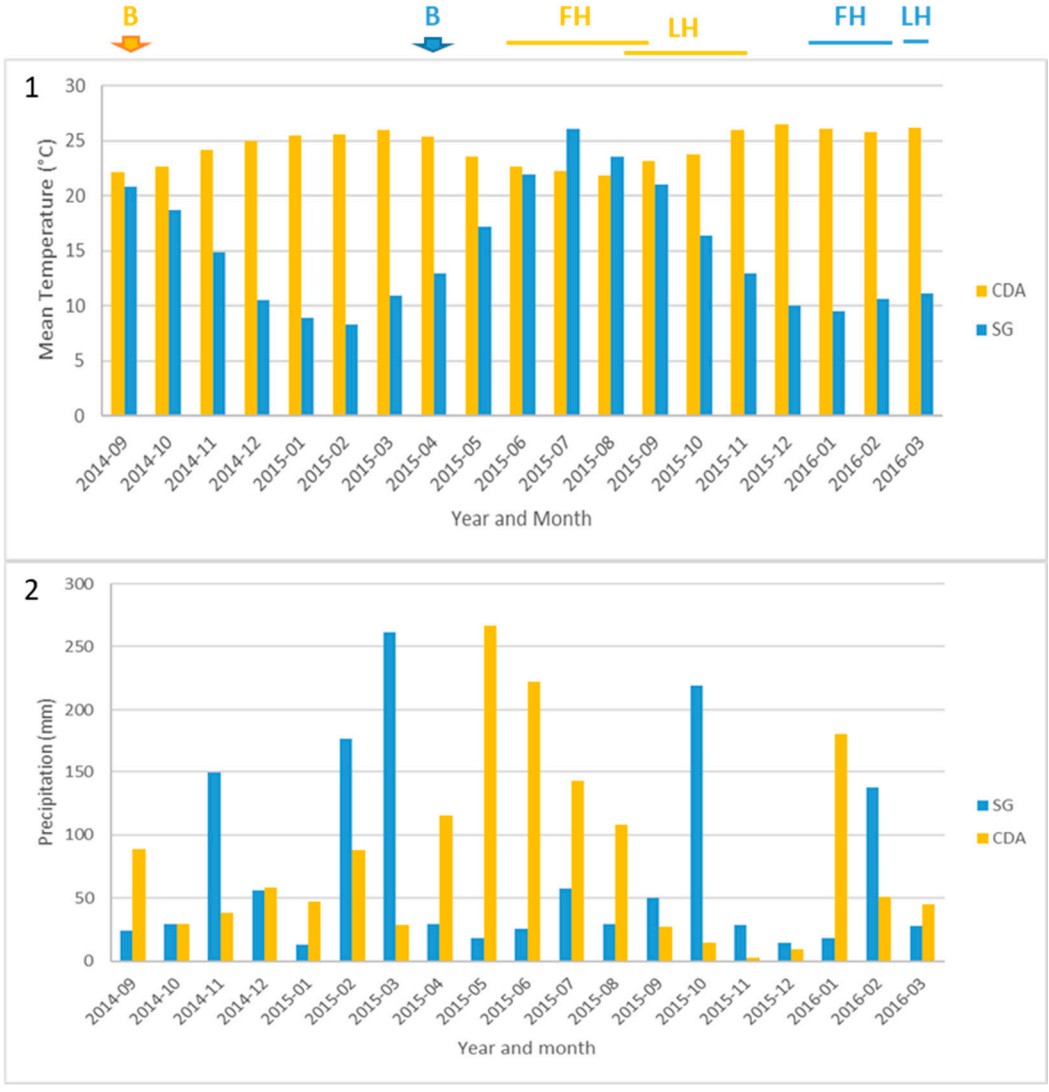

**Figure 1.** Variations in mean monthly temperature (**1**) and precipitation (**2**) at the INRAE-CIRAD research station (SG and in blue) and the EMBRAPA research station (CDA and in yellow) during the year of fruit development and maturation. Blooming (B) as well as fruit and leaf harvesting (FH and LH) periods at each station are indicated at the top of the graphs, in yellow for Cruz das Almas and in blue for San Giuliano.

The soil at the INRAE-CIRAD experimental station is derived from alluvial deposits and classified as fersalitic (pH range 6.0–6.6). The soil in the experimental area of the EMBRAPA research station is a dystrophic cohesive yellow latosol, flat relief, with the following horizons: Ap: 0–0.09 m; AB: 0.09–0.38 m; Bw1: 0.38–0.72 m and Bw2: 0.72–1.20 m, LAd3 (pH range 6.0–6.5) [47].

Different rootstocks were used at the INRAE-CIRAD citrus BRC according to the graft compatibility of each species, while only Rangpur lime was used in the EMBRAPA citrus germplasm collection (Table 1). Trees were planted 6 × 4 m and 7 × 4 m, respectively, in the INRAE-CIRAD and EMBRAPA germplasm collections.

## 2.3. Genetic Conformity of Citrus Cultivars between EMBRAPA and INRAE-CIRAD Germplasm Collections Verified by Molecular Marker Genotyping

Genomic DNA was extracted from leaves sampled from the two germplasm collections using the DNeasy Plant Mini Kit according to the manufacturer's instructions (Qiagen S.A.; USA). The genotyping was conducted with 18 SSR and InDel markers selected according to their distribution on the different genetic linkage groups of the clementine genetic reference map [48–54] (Table 2). PCR was performed as described by [51] in a MWG thermocycler in a final volume of 15 μL containing 10 ng of citrus DNA, 0.2 mM of each dNTP, 10X PCR buffer, (Goldstar, Eurogentec, France) and 1.5 mM $MgCl_2$, 0.2 μM of each primer, and 0.8 u of *Taq* polymerase (Goldstar, Eurogentec, France). The PCR protocol was as follows: denaturation at 94 °C for 5 min followed by 40 repeats of 30 s at 94 °C, 1 min at 50 °C or 55 °C (depending on the melting temperature of the primers), 45 s at 72 °C; and a final elongation step for 4 min at 72 °C.

Amplified fragments were separated on a 6% polyacrylamide (acrylamide/bis-acrylamide19:1) sequencing gel, containing 7 M urea in 0.5 X TBE buffer, at 60 W for 1.5 h to 3 h. Three microliters of PCR product were mixed with an equal volume of loading buffer, containing 95% formamide, 0.25% bromophenol blue, 0.25% xylene cyanol, and 10 mM of EDTA. This mixture was heated for 5 min at 94 °C to denature the DNA before loading. Separated amplified DNA fragments were stained by the silver nitrate method [51].

**Table 2.** Genetic markers used for identity assessment and diversity analysis.

| Scaffold | Marker Id. | SSR Type | Accession Number | Forward Sequence | Reverse Sequence | AT [a] (°C) | Size Range [b] (bp) | Position [c] | Reference |
|---|---|---|---|---|---|---|---|---|---|
| 1 | IDEMA | Indel | | CTCTTTCTGCTTCCTGACATC | GCCGGTGAATAAAACACAAC | 55 | 263–277 | 7406793 | Garcia-Lor et al., 2013 |
| 2 | Ci04H06 | (GA)n | FR677579 | CAAAGTGGTGAAACCTG | GGACATAGTGAGAAGTTGG | 55 | 184–196 | 8097269 | Cuenca et al., 2011 |
| 2 | Ci03C08 | (GA)n | FR677576 | GCTTCTTACATTCCTCAAA | CAGAGACAGCCAAGAGA | 55 | 200–225 | 27339948 | Cuenca et al., 2011 |
| 2 | MEST46 | (CAA)n | DY266484 | GGTGAGCATCTGGACGACTT | GAACCAGAATCAGAACCCGA | 55 | 230–256 | 33532354 | Garcia-Lor et al., 2012 |
| 2 | Ci05A05 | (GA)n | FR677580 | TGGGCTTGTAGACAGTTA | CGGAACAACTAAAACAAT | 50 | 144–179 | 34232309 | Cuenca et al., 2011 |
| 3 | CAC23 | (CAC)n | | TTGCCATTGTAGCATGTTGG | ATCACAATTACTAGCAGCGCC | 55 | 240–260 | 210444 | Kijas et al., 1997 |
| 3 | MEST131 | (GCCCCA)n | DY276912 | GCTGTCACGTTGGGTGTATG | TACCTCCACGTGTCAAACCA | 55 | 120–150 | 50550652 | Garcia-Lor et al., 2012 |
| 4 | Ci03D12a | (GT)n(GA)n | FR677577 | CCCACAACCATCACC | GCCATAAGCCCTTTCT | 50 | 240–280 | 25569961 | Aleza et al., 2011 |
| 5 | MEST88 | (TC)n | DY271576 | ATGAGAGCCAAGAGCACGAT | GCCTGTTTGCTTTCTCTTTCTC | 55 | 99–130 | 36034362 | Garcia-Lor et al., 2012 |
| 6 | MEST192 | (AT)n | DY283129 | CTTGGCACCATCAACACATC | CGCGGATCATCTAGCATACA | 55 | 200–240 | 17474047 | Aleza et al., 2011 |
| 6 | MEST488 | (CT)n | DY297637 | CTTTGCGTGTTTGTGCTGTT | CACGCTCTTGACTTTCTCCC | 55 | 133–164 | 21253670 | Garcia-Lor et al., 2012 |
| 6 | IDPSY | Indel | | CCTGTCGACATTCAGGTTAG | CTCATCACATCTTCGGTCTC | 55 | 246–249 | 21393019 | Garcia-Lor et al., 2013 |
| 6 | Ci01C06 | (CT)n | FR692356 | TGGAGACACAAAGAAGAA | GGACCACAACAAAGACAG | 50 | 131–170 | 24790953 | Cuenca et al., 2011 |
| 7 | MEST107 | (AGA)n | DY274062 | CCCCATCCTTTCAACTTGTG | GCTGAGATGGGGATGAAAGA | 55 | 183–201 | 210493 | Garcia-Lor et al., 2012 |
| 7 | Ci03B07 | (GT)n | FR7677573 | TGAGGGACTAAACAGCA | CACCTTTCCCTTCCA | 55 | 263–279 | 11545443 | Garcia-Lor et al., 2012 |
| 8 | Ci01F04a | (CT)nCC(CT)n | AM489736 | TGCTGCTGCTGTTGTTGTTCT | AAGCATTTAGGGAGGGTCACT | 55 | 190–228 | 1063542 | Froelicher et al., 2008 |
| 8 | Ci02F07 | (GT)n | AJ567406 | TGCTGGTTTTCAGATACTT | GCAGCGTTTGTTTTCT | 55 | 188–215 | 15053136 | Froelicher et al., 2008 |
| 8 | MEST15 | (GAG)n | FC912829 | GCCTCGCATTCTCTTGACTC | TTATTACGAAGCGGAGGTGG | 55 | 192–210 | 24850303 | Garcia-Lor et al., 2012 |

[a] Primer annealing temperature; [b] Range of amplified fragment sizes; [c] Position on the reference citrus genome [55].

### 2.4. Essential Oil Extraction

In the INRAE-CIRAD germplasm, for each cultivar, fruit were collected at the maturity stage in January to February 2016, and leaves in March 2016 (Figure 1). At the EMBRAPA research station, blooming began in September to October 2014 and the fruit were collected at the maturity stage from June to September 2015, according to the citrus cultivar Leaves were collected from August to November 2015.

Fruit and leaves were picked up all around the trees. Two leaf and fruit samples per cultivar (two trees) were obtained in each germplasm collection. For easy peeler fruit, such as mandarins, the rind was peeled manually, while for the other citrus crops with segment adherent peel the flavedo (external part of the fruit peel) was collected using a knife. Fresh materials (200 g) were submitted to water distillation for 3 h using a Clevenger-type apparatus.

### 2.5. Analytical GC

GC analyses were carried out on a Clarus 500 PerkinElmer (Perkin Elmer, Courtaboeuf, France) system equipped with two flame ionization detectors and fused-silica capillary columns (50 m × 0.22 mm i.d., film thickness 0.25 μm), BP-1 (polydimethylsiloxane) and BP-20 (polyethyleneglycol). The oven temperature was programmed from 60 °C to 220 °C at 2 °C/min followed by isothermal hold (20 min); detector temperature, 250 °C; injector temperature, 250 °C (injection mode, split, 1:60); carrier gas, helium (1.0 mL/min). Injected volume: 0.5 μL of a solution of 50 μL of essential oil diluted in 350 μL of $CCl_4$.

### 2.6. GC/MS Analysis

GC/MS analyses were carried out using a Perkin-Elmer TurboMass (Perkin Elmer, Courtaboeuf, France) detector (quadrupole), directly coupled to a Perkin-Elmer Autosystem XL, equipped with a fused-silica capillary column (60 m × 0.22 mm i.d., film thickness 0.25 μm), Rtx-1 (polydimethylsiloxane). Carrier gas, helium at 1 mL/min; split, 1:80; injection volume, 0.2 μL; injector temperature, 250 °C; oven temperature programmed from 60 °C to 230 °C at 2 °C/min followed by isothermal hold (45 min). Ion source temperature, 150 °C; energy ionization, 70 eV; electron ionization mass spectra were acquired over the mass range 35–350 Da.

### 2.7. Component Identification

The component identifications were based:

(a)  on comparison of their GC retention indices (RI) on polar and apolar columns, determined relative to the retention times of a series of *n*-alkanes with linear interpolation with those of authentic compounds and literature data [56];

(b)  on computer matching against commercial mass spectral libraries [57,58] and by comparison of spectra with literature data [56,59,60].

(c)  by $^{13}$C-NMR spectroscopy, following the methodology developed and computerized in our laboratories using a tailored software and spectral data library [61–63].

### 2.8. Data Analysis

Genetic relationships between the different cultivars were analyzed with DARwin software v6 (2019) [64,65] using the factorial analysis method, based on the simple matching similarity index, which took into account the percentage of common alleles between two citrus samples divided by the total number of observed alleles. Chemical data were analyzed using R software v3.6.1 (2019) and the basic packages for calculating means and standard deviations. The Agricolae R package v1.3-1 (2019) was used for the Wilcoxon statistical test of paired analysis of geographical origin at an α risk of 0.05. The Ade4 package v1.7-15 (2013) was used for principal component analysis (PCA) in which the values of each variable were centered and reduced to obtain variations of the same size among

variables. Heat maps were constructed using R software with the g-plots R package v3.0.1.1 (2016) to analyze the EO data and determine the relationships between cultivars and components contributing to this diversity.

## 3. Results

### 3.1. Genetic Diversity of Samples and Conformity of Cultivars between the Two Collections

The citrus accessions with the same denominations present in both collections from INRAE-CIRAD at San Giuliano (Corsica) and EMBRAPA at Cruz das Almas (Bahia) were submitted to genetic analysis using 18 SSR and InDel markers to retain only accessions having the same genetic profile. Accessions with different profiles between the Corsica and Bahia collections were discarded and only 17 cultivars were selected for EO analysis (Table 1). Genotyping data were used to construct a factorial analysis based on dissimilarity indices between each citrus pair to represent the overall organization of the diversity in the citrus family group (Figure 2). The first two axes of the factorial analysis represented 56.79% of the overall diversity. The different cultivars positioned between the four ancestral clusters of citrus genetic diversity (represented in the figure by circles) with a majority of cultivars at or near the mandarin cluster or intermediate between the mandarin and pummelo clusters.

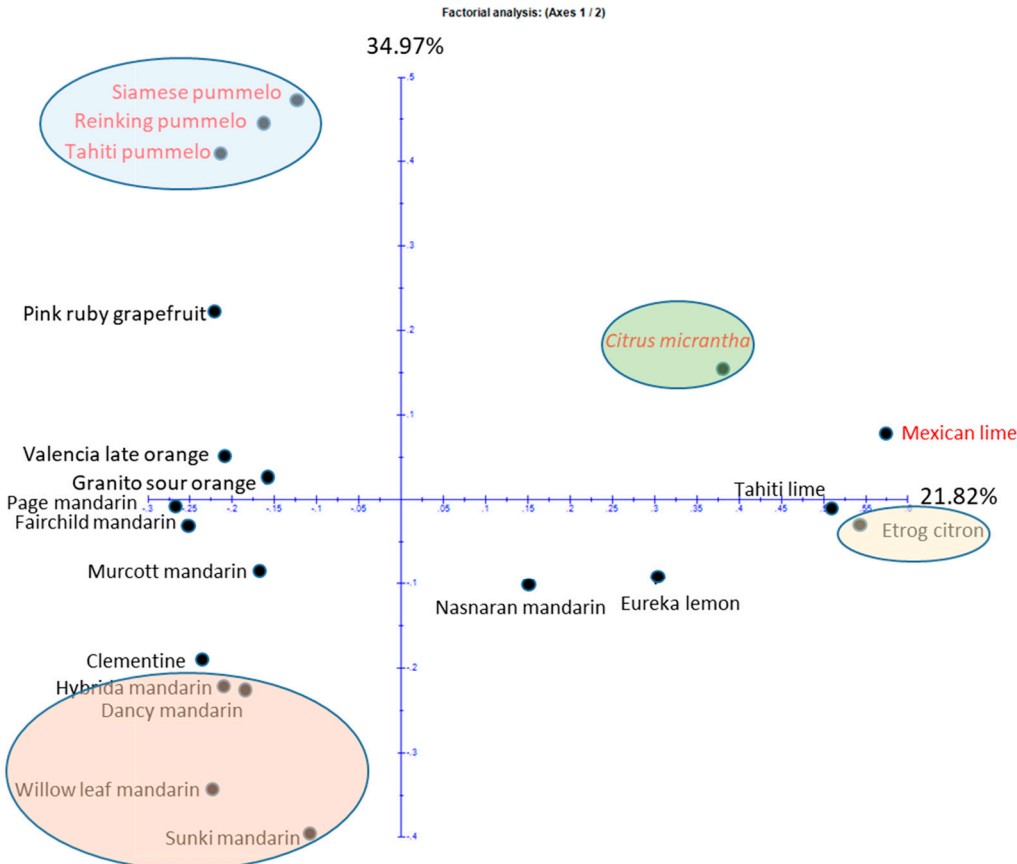

**Figure 2.** Factorial analysis (DFA) of the diversity of the studied citrus cultivars (in black) supplemented citrus species (in red), based on the genetic distance calculated with the allelic data of the 18 markers. The ancestral species are indicated by circles.

### 3.2. Yield of Peel and Leaf Essential Oils

EO yields were generally higher in citrus fruit grown in Corsica than in Bahia (Table 3). The differences were highly significant for peel essential oils (PEOs) (paired *t*-test: $t = 3.11$, $p = 0.0045$) whereas only marginally for leaf essential oils (LEOs) (paired *t*-test: $t = 2.05$, $p = 0.059$).

**Table 3.** Yields (mL/100 g fresh tissue) of EOs extracted from leaf (LEOs) and fruit peel (PEOs) of Corsican and Bahian citrus cultivars.

| Type | Cultivar | PEO | | LEO | |
|---|---|---|---|---|---|
| | | Corsica | Bahia | Corsica | Bahia |
| Citron | Etrog | 0.22 | 0.19 | 0.21 | 0.12 |
| Lemon | Feminello | 1.00 | 0.29 | 0.32 | 0.35 |
| Lemon | Eureka | 1.00 | 0.20 | 0.30 | ND |
| Lime | Tahiti | 0.63 | 0.16 | 0.27 | 0.10 |
| Clementine | Tomatera | 0.32 | 0.35 | 0.33 | 0.13 |
| Mandarin | Dancy | 0.61 | 0.26 | 0.35 | 0.34 |
| Mandarin | Fairchild | 0.28 | 0.47 | 0.10 | 0.15 |
| Mandarin | Willowleaf | 0.50 | 0.41 | 0.29 | 0.23 |
| Mandarin | Murcott | 0.57 | 0.50 | 0.20 | 0.16 |
| Mandarin | Sunki | 0.30 | 0.09 | 0.15 | 0.15 |
| Mandarin | Page | 0.25 | 0.25 | 0.18 | 0.10 |
| Mandarin | Hybrida | 0.45 | 0.25 | ND | ND |
| Mandarin | Nasnaran | 1.00 | 0.48 | 0.30 | 0.12 |
| Sour orange | Granito | 0.45 | 0.47 | 0.24 | 0.23 |
| Orange | Hamlin | 1.45 | 1.11 | 0.15 | 0.12 |
| Orange | Valencia late | 0.38 | 0.33 | 0.15 | 0.10 |
| Grapefruit | Pink ruby | 0.20 | 0.21 | 0.10 | 0.23 |
| Total set * | | 0.57 ± 0.35 | 0.35 ± 0.23 | 0.23 ± 0.08 | 0.18 ± 0.08 |
| Citron family * | | 0.71 ± 0.37 | 0.21 ± 0.06 | 0.28 ± 0.05 | 0.19 ± 0.14 |
| Mandarin family * | | 0.48 ± 0.24 | 0.34 ± 0.14 | 0.24 ± 0.09 | 0.17 ± 0.08 |

ND: Not determined; * Mean value ± Standard deviation.

The greatest differences were observed in the citron-related group, including lemons and lime, with an increase of about 3.5-fold of PEOs for the Corsican cultivars, with the highest yield difference (x4) observed in lemons. Although these differences were lower for the overall mandarin group, the yields of PEOs of some cultivars such as "Dancy", "Sunki" and "Nasnaran" were 2- to 3-fold higher in Corsica than in Brazil. It should be noted that "Tahiti" lime and "Fairchild" mandarin produced a 2-fold higher quantity of EOs in the peel of fruit produced in Bahia than in Corsica. For other citrus fruit (orange, sour orange and grapefruit) yields were equivalent between the two geographical sites. Variations in EO yields were also observed between citrus cultivars. Regarding PEOs, the highest yield was obtained for "Hamlin" orange (1.45 mL/kg fruit peel) in Corsica and in Brazil, and the lowest yield was obtained for "Sunki" mandarin in Bahia (0.09 mL/kg fruit peel) and grapefruit in Corsica (0.20 mL/kg fruit peel).

The differences between LEO yields of Corsican and Brazilian citrus were not significant at $\alpha$ risk equal to 0.05. LEO yields were more homogeneous than PEO yields and ranged from 0.10 to 0.35 mL/kg in both Corsican or Brazilian samples. The species or groups of citrus types did not appear to be distinguished by a specific yield, except for the lemon LEO yield which was rather high while that of orange was rather low regardless of the provenance. The yields within mandarins were highly variable.

### 3.3. Composition of Essential Oils

#### 3.3.1. Leaf Essential Oils

A total of 71 compounds constituted 87.2 to 99.6% of the overall composition of Brazilian citrus LEOs while 62 compounds constituted 93.7 to 99.7% of the overall composition of Corsican citrus LEOs. Twenty-seven major compounds with a proportion higher, than 1% in at least one cultivar, were observed (Table 4).

**Table 4.** Mean proportion (%) and standard deviation (SD) of major LEO compounds for all citrus cultivars, and mandarin and citron families. The values on a gray background have significant differences according to the Wilcoxon test (*p* value < 0.05). Compounds are listed in increasing order of their retention indices (RI) on the apolar column.

| LEOs | | All Cultivars | | | | Mandarin Family | | | | Citron Family | | | |
| --- | --- | --- | --- | --- | --- | --- | --- | --- | --- | --- | --- | --- | --- |
| | | Bahia | | Corsica | | Bahia | | Corsica | | Bahia | | Corsica | |
| RI | Compounds | Mean | SD | Mean | SD | Mean | SD | Mean | SD | Mean | SD | Mean | SD |
| 929 | α–pinene | 0.96 | 0.89 | 1.19 | 0.87 | 1.29 | 0.80 | 1.61 | 0.94 | 0.20 | 0.06 | 0.66 | 0.37 |
| 963 | sabinene | 13.75 | 20.12 | 12.46 | 15.32 | 16.85 | 18.29 | 16.97 | 17.06 | 0.75 | 0.36 | 2.75 | 1.07 |
| 970 | β pinene | 5.60 | 12.62 | 8.95 | 15.06 | 7.42 | 16.00 | 10.41 | 20.04 | 2.74 | 1.90 | 8.80 | 8.61 |
| 979 | myrcene | 1.14 | 0.88 | 1.68 | 1.15 | 0.97 | 0.84 | 1.69 | 1.20 | 0.78 | 0.39 | 0.98 | 0.05 |
| 1007 | 3-carene | 1.27 | 2.39 | 0.82 | 1.23 | 1.14 | 2.52 | 0.89 | 1.53 | 0.00 | 0.00 | 0.33 | 0.09 |
| 1010 | α–terpinene | 0.16 | 0.17 | 1.04 | 1.12 | 0.15 | 0.13 | 1.30 | 1.45 | 0.00 | 0.00 | 0.56 | 0.40 |
| 1011 | p-cymene | 1.94 | 2.09 | 0.77 | 0.87 | 2.45 | 1.33 | 0.85 | 1.23 | 0.39 | 0.15 | 0.65 | 0.01 |
| 1020 | limonene | 10.63 | 13.81 | 13.84 | 12.20 | 3.60 | 2.72 | 5.82 | 2.42 | 25.95 | 14.76 | 29.49 | 9.30 |
| 1020 | β-phellandrene | 0.53 | 0.84 | 0.55 | 0.50 | 2.43 | 6.53 | 0.34 | 0.34 | 1.25 | 1.29 | 0.66 | 0.43 |
| 1035 | (E)-β-ocimene | 0.99 | 1.54 | 4.30 | 3.20 | 1.10 | 1.77 | 5.33 | 3.24 | 0.84 | 1.13 | 1.70 | 0.40 |
| 1046 | γ-terpinene | 1.83 | 4.26 | 3.34 | 4.03 | 4.58 | 6.91 | 5.26 | 4.54 | 0.28 | 0.04 | 0.44 | 0.10 |
| 1070 | p-cymenene | 0.23 | 0.33 | 0.01 | 0.03 | 0.38 | 0.42 | 0.00 | 0.00 | 0.03 | 0.05 | 0.00 | 0.00 |
| 1082 | linalool | 7.02 | 19.41 | 5.40 | 15.54 | 19.92 | 22.47 | 15.20 | 16.07 | 1.53 | 10.11 | 2.05 | 0.79 |
| 1129 | citronellal | 7.58 | 19.57 | 4.60 | 14.01 | 9.29 | 24.21 | 8.49 | 21.39 | 3.55 | 3.56 | 2.22 | 1.09 |
| 1161 | terpinen-4-ol | 1.46 | 1.83 | 2.55 | 2.97 | 1.44 | 1.58 | 3.14 | 2.71 | 0.43 | 0.09 | 0.54 | 0.27 |
| 1170 | α-terpineol | 1.07 | 2.10 | 1.27 | 2.50 | 0.56 | 0.66 | 0.51 | 0.42 | 0.43 | 0.27 | 0.79 | 0.36 |
| 1208 | nerol | 0.70 | 1.15 | 0.85 | 1.25 | 0.00 | 0.00 | 0.01 | 0.03 | 1.93 | 1.42 | 1.86 | 1.71 |
| 1213 | neral | 6.35 | 10.72 | 4.35 | 6.61 | 0.03 | 0.10 | 0.09 | 0.14 | 20.48 | 8.67 | 13.77 | 2.08 |
| 1213 | thymyl methyl oxide | 0.50 | 1.35 | 0.83 | 2.47 | 0.66 | 1.67 | 1.55 | 3.31 | 0.00 | 0.00 | 0.00 | 0.00 |
| 1232 | geraniol | 0.85 | 1.30 | 0.76 | 1.48 | 0.07 | 0.23 | 0.02 | 0.03 | 1.37 | 1.01 | 0.68 | 0.49 |
| 1242 | geranial | 10.18 | 15.54 | 8.25 | 11.12 | 0.01 | 0.03 | 0.19 | 0.22 | 24.80 | 15.36 | 18.00 | 2.91 |
| 1244 | linalyl acetate | 2.46 | 8.55 | 1.90 | 7.85 | 0.07 | 0.17 | 0.00 | 0.00 | 0.15 | 0.29 | 0.00 | 0.00 |
| 1261 | thymol | 0.72 | 2.22 | 0.69 | 2.11 | 1.04 | 2.84 | 1.28 | 2.83 | 0.11 | 0.24 | 0.00 | 0.00 |
| 1341 | neryl acetate | 1.34 | 1.41 | 2.32 | 2.76 | 0.05 | 0.15 | 0.32 | 0.32 | 2.37 | 2.16 | 7.19 | 2.69 |
| 1359 | geranyl acetate | 1.35 | 1.65 | 1.63 | 2.22 | 0.19 | 0.33 | 0.22 | 0.13 | 2.21 | 1.06 | 3.42 | 2.34 |
| 1380 | Me N-methylanthranilate | 5.30 | 19.02 | 5.06 | 18.24 | 17.16 | 30.88 | 9.39 | 24.85 | 0.00 | 0.00 | 0.00 | 0.00 |
| 1673 | β-sinensal | 0.10 | 0.33 | 0.57 | 0.77 | 0.22 | 0.53 | 0.53 | 0.79 | 0.00 | 0.00 | 0.00 | 0.00 |
| 1725 | α-sinensal | 0.06 | 0.13 | 0.43 | 0.39 | 0.14 | 0.18 | 0.55 | 0.68 | 0.00 | 0.00 | 0.00 | 0.00 |

The EO chemical composition between the two geographical sites varied significantly for 6 compounds in the overall sample (γ-terpinene, β-sinensal, α-sinensal, (E)-β-ocimene, p-cymenene), 3 for the mandarin family ((E)-β-ocimene, p-cymenene, geranial) and 5 for the citron family (α-pinene, sabinene, 3-carene, α-terpinene, neryl acetate). All of these compounds were detected in higher proportions in LEOs from Corsica than in those from Brazil (Figure 3). β-sinensal and α-sinensal were mainly present in sour orange, grapefruit and orange LEOs.

In order to reveal possible signatures of biochemical profiles linked to an environmental effect on specific genetic groups, the cultivars were arranged in the heat maps according to the results of the previous genetic diversity analysis. The proportion of each compound was centered and reduced according to the proportions from the two geographical locations (Figure 3). This representation highlights the main differences in both geographical cultivation locations of the same citrus cultivars, and the diversity structure of the compound that supports this varietal diversity organization. The most striking geographical markers were octanal and isogeranial for citron and "Tahiti" lime, neryl acetate for lemons, p-cymene for orange and clementine, p-cymenene for "Dancy" mandarin, 1-8, cineole for "Sunki" mandarin and citronellol for "Nasnaran" mandarin. The chemical signature was highly distinct in mandarin hybrids such as orange, clementine, "Page" and "Murcott", with higher proportions found in Corsican hybrids for a specific group of compounds (δ-elemene, β-sinensal, sabinene, myrcene, terpinen-4-ol, terpinolene, α-sinensal, α-terpinene and (E)-β-ocimene) (Figure 3: dotted frames). "Granito" sour orange was the only citrus fruit with a very similar chemical profile at the two geographical cultivation locations.

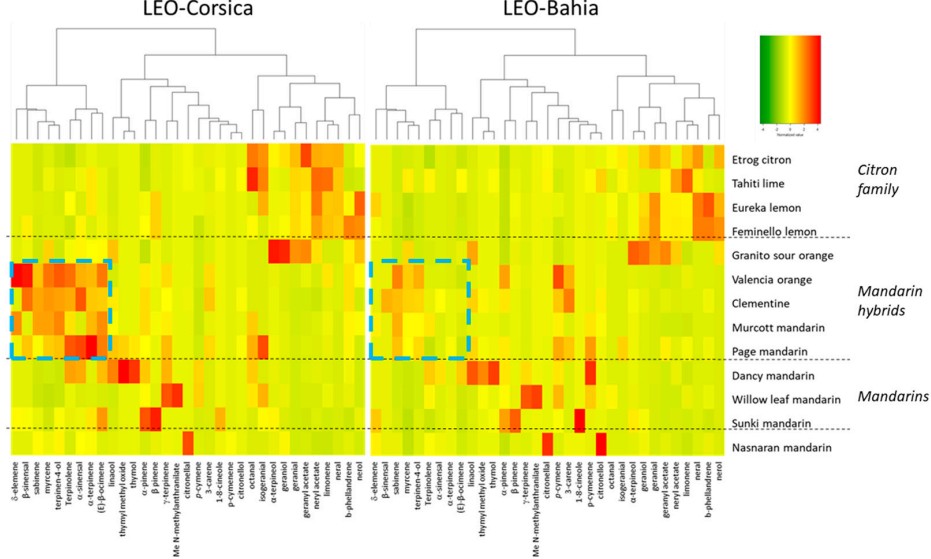

**Figure 3.** Heatmap of citrus chemical diversity and the relationship based on standardized values for the proportions of LEO components between 13 cultivars cultivated in Corsica (left) or Bahia (right). A blue dotted line frames the area of the main variations in compounds between the two locations for the hybrid mandarin cultivars.

To highlight the environmental effect, and considering the high phenotypic variability between the mandarin and citron families, multivariate analyses were carried out by separating the two genetic groups, i.e., the mandarin family and the citron family (or acidic citrus fruit). Orange, sour orange and "Nasnaran" were removed from the mandarin family because they bear a high proportion of the genome of the *C. maxima* ancestral species (for orange and sour orange), and of the *C. micrantha* genome (for "Nasnaran").

The better distinction between geographical areas was revealed by the combination of PCA axes 1 and 3 (Figure 4). The inertia of axis 3 (16.1%) was very close to that of axis 2 (16.9%) and the sum of the two axes represented 40.7% of the total inertia variance. When taking the geographical location

into account, the cultivars cultivated in Corsica differed chemically from the same cultivars cultivated in Brazil (Figure 4—colored circles). This distinction was mainly based on 5 compounds, i.e., (E)-β-ocimene, α-terpinene p-cymene, β-pinene, myrcene, and β-phellandrene. The main exception concerned "Willowleaf" mandarin cultivated in Corsica, with a position close to the cultivars cultivated in Bahia. The chemical composition of "Sunki" mandarin (from both Corsica and Brazil) appeared to be different from that of other studied mandarin cultivars, with a high proportion of α- and β-pinene and 1-8-cineole. Interestingly, chemical differences were noted in "Sunki" mandarins cultivated in Corsica and Bahia.

The effect of the environment on the LEO composition was most evident in the citron family group (citron, lime and lemons) (Figure 5). The chemical composition of cultivars cultivated in Corsica differed greatly from the same cultivars cultivated in Brazil, mainly with regard to 2 compounds: α-pinene and sabinene (axis 2). All of the acidic cultivars cultivated in Corsica had higher proportions of these two monoterpenes than those cultivated in Bahia. It should be noted that 3-carene and α-terpinene were not present in leaves of the acidic cultivars cultivated in Bahia (Table 4).

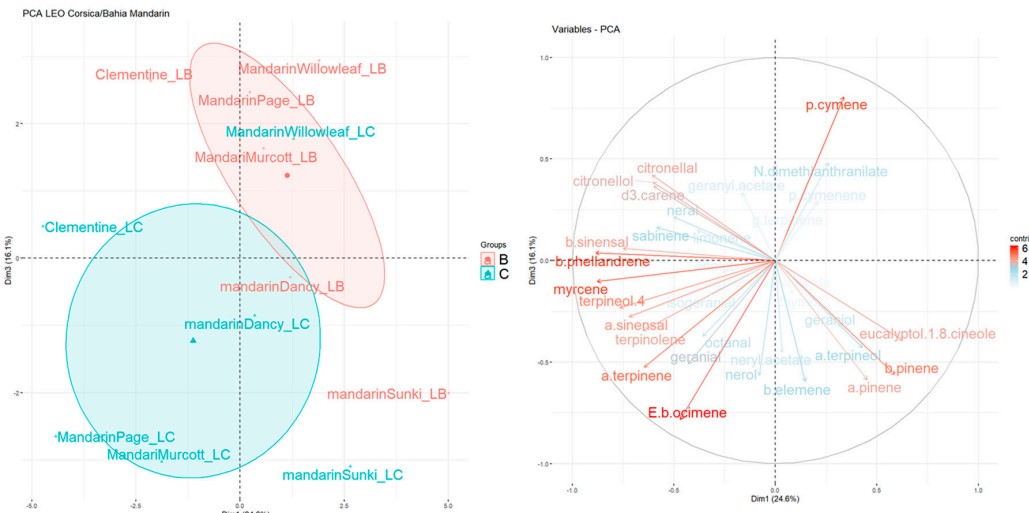

**Figure 4.** PCA of the chemical diversity of the mandarin group based on LEO components (**left**) and the contribution of each compound to the diversity of the considered cultivars (**right**). The geographical location is distinguished by different colors and letter codes, i.e., blue and LC for Corsica and red and LB for Bahia.

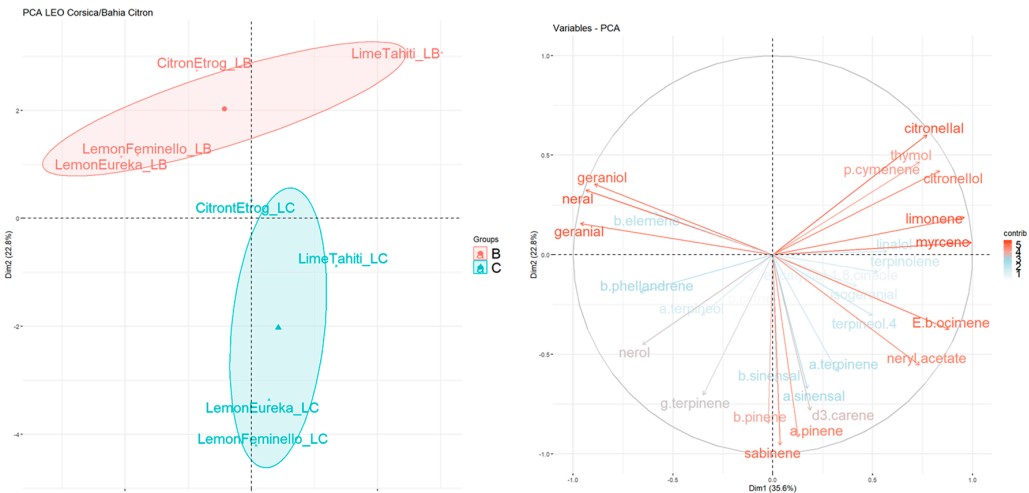

**Figure 5.** PCA of the chemical diversity of the citron-related citrus cultivars based on LEO components (**left**) and the contribution of each compound to the diversity of the considered cultivars (**right**). The geographical location is distinguished by colors and letter codes, i.e., blue and LC for Corsica and red and LB for Bahia.

### 3.3.2. Peel Essential Oils

A total of 67 compounds represented 97.6 to 100% of the overall composition of the PEOs from Bahia and 64 compounds constituted 93.7 to 99.1% of the PEOs from Corsica. Only 22 compounds had proportions greater than 1% in at least one citrus composition (Table 5). This number of compounds was lower than that of LEOs and this was related to the dominance of limonene, which on average constituted 75.6% of the PEOs (ranging from 43.5 to 89%), compared to an average of 15.2% for LEOs (range: 3.6 to 32%).

Heat maps were drawn up using the same process as for LEOs (Figure 6). The most remarkable differences were observed in citron and Tahiti lime, with a first group of 5 compounds (geranyl acetate, isogeranial, neryl acetate, neral, geranial) found in greater proportions in PEOs from Corsica. For citron cultivated in Corsica, 3 other compounds (nerol, geraniol and α-terpinene) were also detected in higher proportions, while in Bahia, p-cymene, thymol and α-pinene were in higher proportions. Geranyl acetate was also dominant in Corsican lemons. As with the LEOs, markers of the geographical cultivation location could also be mentioned for other cultivars: octanal for grapefruit, 3-carene for "Valencia" orange and clementine, linalool for "Dancy" mandarin, thymyl methyl oxide for "Page" mandarin, α-sinensal for "Fairchild" mandarin, Me N-methylanthranilate for "Willowleaf" mandarin, terpinolene for "Sunki" mandarin and geraniol for "Nasnaran" mandarin.

When only taking major compounds (>1%) into account, chemical profiles of grapefruit, sour orange and sweet orange were very similar regardless of the cultivation location.

Principal component analysis (PCA) performed to highlight the overall effect of the environment on citron the PEO chemical composition showed a geographical effect of cultivation (Supplementary File 1). This geographical chemical distinction between Corsica and Bahia was based on 2 cultivars ("Etrog" citron and "Tahiti" lime) and 7 components: myrcene, limonene, citronellal, citronellol, geraniol, geranial and neral (axis 2). With the exception of myrcene and limonene, all of these compounds were found in greater proportions in cultivars cultivated in Corsica, whereas limonene was detected in a lesser proportion.

The β-pinene content was higher in PEOs from Corsica for Tahiti lime and Eureka lemon (10.27% on average in Corsica versus 2.05% in Bahia). The PEO chemical profiles of citron "Etrog" differed markedly depending on the cultivation location. The proportions of cymene, thymol and α-pinene were higher in PEOs from Bahia (average values of 7.26% in Bahia compared to 1.71% in Corsica), while the levels of neral, geranial, nerol, geraniol, neryl acetate, geranyl acetate and α-terpineol were higher in PEOs from Corsica (average values of 3.20% in Corsica compared to 0.15% in Bahia). These differences according to the cultivation location for these 7 latter compounds were also observed with regard to PEOs of "Tahiti" lime, with greater proportions detected in PEOs of citrus trees cultivated in Corsica. Note that the effect of the environment was the opposite concerning the proportions of geranial and geraniol in LEOs of "Feminello" and "Eureka" lemons, with 35% and 1.8% respectively when cultivated in Bahia compared to only 15% and 0.5% in Corsica. The greatest differences due to geographical origin were observed with γ-terpinene for "Tahiti" lime, i.e., the proportion was 17% for EOs from Corsican cultivars, while it represented only 3.3% of the total EO composition in citrus cultivars produced in Bahia.

In lime and lemons, the average amount of limonene was 78.2% in PEOs produced in Brazil and 53% in those produced in Corsica.

With the mandarin group there was no clear distinction between the two geographical origins (Supplementary File 2). The differences were, however, greater for individual cultivars, such as "Dancy", "Hybrida" and "Fairchild" mandarins.

**Table 5.** Mean (%) and standard deviation (SD) of major compound proportions of PEOs of all citrus cultivars, and the mandarin and citron families. The values on a light grey background are statistically different between the two cultivation sites (Wilcoxon test *p* value < 0.05). Compounds are listed in increasing order of their retention indices (RI) on the apolar column.

| | PEOs | All Cultivars | | | | Mandarin Family | | | | Citron Family | | | |
| | | Corsica | | Bahia | | Corsica | | Bahia | | Corsica | | Bahia | |
| RI | Compounds | Mean | SD | Mean | SD | Mean | SD | Mean | SD | MEAN | SD | Mean | SD |
|---|---|---|---|---|---|---|---|---|---|---|---|---|---|
| 929 | α-pinene | 0.65 | 0.50 | 0.82 | 0.59 | 0.57 | 0.35 | 0.61 | 0.30 | 0.94 | 0.71 | 1.55 | 0.95 |
| 963 | sabinene | 0.75 | 0.99 | 0.71 | 0.99 | 0.36 | 0.25 | 0.41 | 0.34 | 0.94 | 0.49 | 0.93 | 0.58 |
| 970 | β-pinene | 3.14 | 4.46 | 2.49 | 3.39 | 0.75 | 0.68 | 1.47 | 1.77 | 7.51 | 4.74 | 4.25 | 3.30 |
| 977 | octanal | 0.10 | 0.19 | 0.45 | 0.49 | 0.16 | 0.23 | 0.51 | 0.47 | 0.00 | 0.00 | 0.09 | 0.16 |
| 979 | myrcene | 1.45 | 0.23 | 1.60 | 0.14 | 1.47 | 0.13 | 1.64 | 0.08 | 1.18 | 0.20 | 1.47 | 0.06 |
| 1010 | α-terpinene | 0.32 | 0.26 | 0.21 | 0.23 | 0.22 | 0.19 | 0.22 | 0.22 | 0.42 | 0.29 | 0.26 | 0.21 |
| 1011 | *p*-cymene | 0.78 | 1.28 | 1.87 | 4.08 | 0.29 | 0.49 | 0.50 | 1.01 | 2.51 | 1.43 | 6.22 | 6.82 |
| 1020 | limonene | 79.45 | 26.14 | 83.71 | 12.65 | 89.01 | 8.85 | 88.92 | 6.12 | 54.61 | 13.27 | 68.52 | 10.44 |
| 1046 | γ-terpinene | 4.02 | 6.11 | 2.71 | 3.84 | 3.24 | 5.66 | 2.47 | 4.18 | 9.96 | 2.67 | 6.04 | 4.28 |
| 1077 | terpinolene | 0.37 | 0.34 | 0.26 | 0.20 | 0.29 | 0.32 | 0.24 | 0.18 | 0.62 | 0.37 | 0.28 | 0.10 |
| 1082 | linalool | 1.47 | 1.90 | 0.87 | 0.85 | 2.25 | 2.85 | 1.16 | 1.16 | 0.79 | 0.53 | 0.47 | 0.75 |
| 1129 | citronellal | 0.68 | 0.98 | 0.53 | 0.74 | 0.63 | 0.32 | 0.38 | 0.26 | 0.42 | 0.28 | 0.28 | 0.55 |
| 1161 | terpinen-4-ol | 0.77 | 1.15 | 0.45 | 0.51 | 0.24 | 0.14 | 0.39 | 0.26 | 0.99 | 0.84 | 0.36 | 0.24 |
| 1170 | α-terpineol | 1.11 | 1.55 | 0.43 | 0.38 | 0.45 | 0.33 | 0.33 | 0.17 | 2.23 | 2.34 | 0.55 | 0.45 |
| 1213 | neral | 1.19 | 1.70 | 0.39 | 0.46 | 0.38 | 0.27 | 0.24 | 0.35 | 3.68 | 1.86 | 0.91 | 0.59 |
| 1232 | geraniol | 0.26 | 0.60 | 0.05 | 0.08 | 0.09 | 0.23 | 0.04 | 0.08 | 0.65 | 1.01 | 0.09 | 0.10 |
| 1242 | geranial | 1.48 | 2.34 | 0.43 | 0.21 | 0.36 | 0.28 | 0.28 | 0.27 | 4.82 | 2.70 | 1.03 | 1.15 |
| 1341 | neryl acetate | 0.44 | 0.55 | 0.14 | 0.14 | 0.14 | 0.14 | 0.07 | 0.12 | 1.14 | 0.53 | 0.22 | 0.14 |
| 1359 | geranyl acetate | 0.43 | 0.43 | 0.20 | 0.23 | 0.14 | 0.14 | 0.17 | 0.25 | 0.96 | 0.35 | 0.24 | 0.27 |
| 1380 | Me *N*-methylanthranilate | 0.08 | 0.27 | 0.03 | 0.10 | 0.19 | 0.41 | 0.05 | 0.13 | 0.00 | 0.00 | 0.04 | 0.09 |
| 1431 | trans-α-bergamotene | 0.22 | 0.39 | 0.12 | 0.17 | 0.00 | 0.00 | 0.00 | 0.00 | 0.60 | 0.48 | 0.32 | 0.14 |
| 1495 | β-bisabolene | 0.18 | 0.42 | 0.11 | 0.24 | 0.00 | 0.00 | 0.00 | 0.00 | 0.61 | 0.61 | 0.39 | 0.30 |

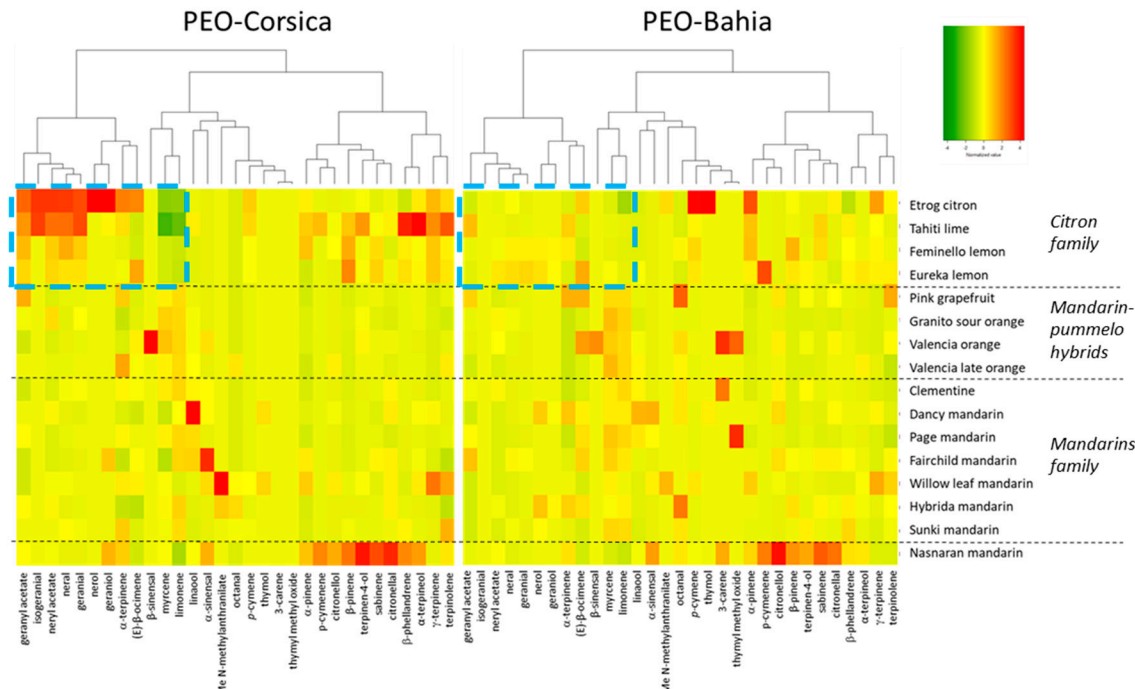

**Figure 6.** Heatmap of citrus chemical diversity and the relationship based on standardized values for the proportions of PEO components between 16 cultivars cultivated in Corsica (left) or Bahia (right). A blue dotted line frames the area of the main variations in compounds between the two locations for the citron family cultivars.

## 3.4. Site-Specific Compounds

Twenty-four compounds were detected from only one cultivation location (7 in Corsica and 17 in Bahia), including 19 in PEOs and 9 in LEOs (Table 6). These compounds were essentially found in EOs of the citron family cultivated in Corsica, with 7 specific compounds, whereas in Bahia specific compounds were detected all the cultivars. The specific compounds most frequently encountered in EOs from trees cultivated in Bahia were octyl acetate (in 6 PEOs and 8 LEOs) and undecanal in Corsica (detected in 5 cultivars but only in LEOs). Apart from carvone (1.6% in EOs of "Etrog" citron cultivated in Bahia), none of these compounds had a content of over 1%.

**Table 6.** Proportion of fruit peel and LEO compounds specific to one site for each cultivar.

| IK apol | Compounds | Site | Etrog citron | Tahiti lime | Lemon Feminello | Lemon Eureka | Grapefruit Pink | Sour orange Granito | Orange Hamlin | Orange Valencia late | Mandarin Willowleaf | Mandarin Dancy | Mandarin Page | Mandarin Fairchild | Mandarin Sunki | Mandarin Hybrida | Clementine | Tangor Murcott | Mandarin Nasnaran |
|---|---|---|---|---|---|---|---|---|---|---|---|---|---|---|---|---|---|---|---|
| | **PEOs** | | | | | | | | | | | | | | | | | | |
| 958 | 6-methyl-hept-5-en-2-one | Bahia | 0.9 | | 0.1 | | | | | | | | | | | | | | |
| 1074 | trans linalool oxide THF form | Bahia | | | | | | | | | | | | | | | | | 0.2 |
| 1143 | p-mentha-1,5-dien-8-ol | Corsica | | 0.9 | | | | | | | | | | | | | | | |
| 1146 | isoneral | Corsica | 0.1 | | | | | | | | | | | | | | | | |
| 1147 | borneol | Corsica | | 0.2 | | | | | | | | | | | | | | | |
| 1157 | p-cymen-8-ol | Corsica | | 0.3 | | | | | | | | | | | | | | | |
| 1192 | octyl acetate | Bahia | | | | | 0.7 | 0.3 | 0.8 | 0.2 | | | | 0.4 | | | | | 0.1 |
| 1196 | trans-carveol | Bahia | | | | | | | | | | | | 0.3 | | | 0.4 | | 0.1 |
| 1222 | carvone | Bahia | 1.6 | | | | | | | 0.1 | | | | | | | | 0.7 | |
| 1332 | α-terpinyl acetate | Bahia | | | | | | | | | | | 0.3 | | | | | | 0.4 |
| 1333 | citronellyl acetate | Bahia | | | | | | | 0.2 | | | | | 0.7 | | | | | 0.2 |
| 1334 | δ–elemene | Bahia | | | | | | | | | | 0.2 | | | 0.4 | | | | |
| 1375 | α-copaene | Bahia | 0.3 | | | | | | | | 0.8 | | | 0.4 | | | | | 0.4 |
| 1479 | β-selinene | Corsica | | | | | | | | 0.4 | | | | | | | | | |
| 1480 | germacrene D | Bahia | | | | | | | | | | 0.2 | | | | 0.6 | | | 0.2 |
| 1489 | bicyclogermacrene | Bahia | 0.3 | | | | 0.2 | | | | | | | 0.2 | 0.6 | | | | |
| 1572 | caryophyllene oxide | Bahia | | | | | | | | | | | | 0.7 | 0.1 | | | | |
| 1624 | τ-cadinol | Bahia | 0.1 | | | 0.1 | | | | | | | | | | | | | |
| 1772 | nootkatone | Corsica | | | | | 0.3 | | | | | | | | | | | | |
| | **LEOs** | | | | | | | | | | | | | | | | | | |
| 1051 | octanol | Corsica | | 0.3 | | | | | | | | | | | | | | | |
| 1074 | trans linalool oxide THF form | Bahia | | | | | | | | | | | 0.7 | | 0.2 | | 0.6 | | 0.6 |
| 1125 | cis-verbenol | Bahia | | | | | | | | | | | 0.2 | | | | | | |
| 1179 | myrtenol | Bahia | | | | | 0.2 | | | | | | | 0.6 | | | | | |
| 1192 | octyl acetate | Bahia | | 0.1 | | | 0.5 | | | 0.5 | | | 0.5 | 0.1 | 0.5 | | | 0.2 | 0.2 |
| 1196 | trans-carveol | Bahia | | | | | | | | | | | 0.5 | 0.3 | 0.5 | | 0.2 | 0.6 | |
| 1222 | carvone | Bahia | | | | | | | | 0.3 | | | | | 0.5 | | 0.2 | | |
| 1267 | bornyl acetate | Bahia | | | | | | | | | | | | | | | | | |
| 1284 | undecanal | Corsica | 0.2 | 0.2 | | 0.2 | | | | | | | | | | | | 0.2 | |
| 1532 | β-elemol | Bahia | | | | | 0.2 | | | 0.2 | | | | | 0.5 | | | 0.2 | |

*3.5. EO Composition Stability over Time and Temperature*

To study the stability of the EO profiles over time, we compared data obtained from 2016/2017 extractions with those from the 1996–1999 period [66], including the climate data from these two periods (Table 7).

**Table 7.** Average temperatures per month and per period of fruit development phases during the 1996/1999 and 2015/2016 fruiting seasons (Phases I, II, III correspond to "cell multiplication"; "growing", "ripening", according to [67]).

| Year | Month | | | | | | | | | Fruiting Phases | | |
|---|---|---|---|---|---|---|---|---|---|---|---|---|
| | June | July | August | September | October | November | December | January | February | I + II | III | All |
| 2016/2017 | 21.9 | 26.1 | 23.6 | 21.0 | 16.4 | 12.9 | 10 | 9.6 | 10.6 | 23.8 | 13.4 | 16.9 |
| 1998/1999 | 21.5 | 24.9 | 23.5 | 20.5 | 16.3 | 11.3 | 8.7 | 9.1 | 8.1 | 23.4 | 12.3 | 16.0 |
| 1997/1998 | 21.3 | 23.1 | 24.1 | 21.8 | 17.2 | 12.7 | 10.8 | 9.8 | 10.3 | 22.9 | 13.8 | 16.5 |
| 1996/1997 | 21.1 | 23.4 | 23.4 | 18.4 | 15.8 | 12.9 | 9.8 | 10.3 | 10.7 | 22.6 | 13 | 15.9 |

Temperature was the only factor that varied between the two study periods, i.e., 1996–1999 and 2016/2017. Over the entire period from fruiting to EO extraction, the 2016/2017 season had average temperatures that were 0.4 to 1 °C higher than during the 3 seasons of the 1990s. This difference was also noted over the summer period, corresponding to phases I and II of fruit development in Corsica, with a maximum difference of 1.2 °C with regard to the 1996/1997 season. Rainfall also differed between the two study periods, but these differences were offset by irrigation. Over the 20-year interval between the two studies, no significant differences were observed in the composition of the PEOs, with the exception of "Etrog" citron, whose EO composition differed mainly in the proportion of γ-terpinene, i.e., 30.7% in 1999 and 9.7% in 2016 (Figure 7). In LEOs, on the other hand, differences were observed between the two periods only in "Sunki" mandarin and in the citron family (Figure 7). Some compounds were found in different proportions in LEOs of the citron family between the 2016 ("16") and 1999 ("99") analyses, such as nerol (3.6% vs. 0.4%), 6-methylhept-5-en-2-one (3.3% vs. 0.4%), geraniol (3.1% vs. 0.5%), neryl acetate (3.2% vs. 7.7%) and citronellal (1.5% vs. 2.5%). The chemical composition of "Sunki" mandarin seems to have changed between 1999 and 2016 with, in particular, a variation in the β-pinene content (4.2% in 1999; 55.8% in 2016).

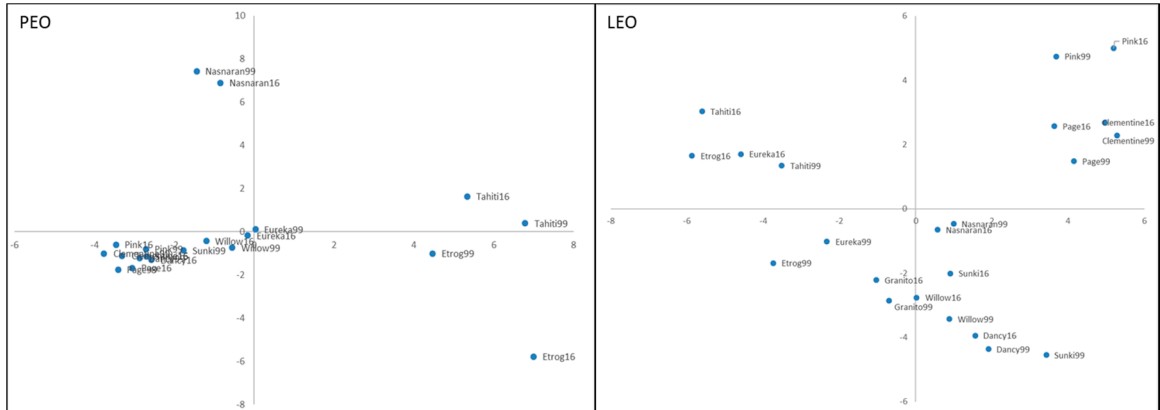

**Figure 7.** PCA of compositions of citrus LEOs and PEOs obtained from the same species and/or cultivars harvested in INRAE-CIRAD citrus germplasm in 2016 (present work) and in 1999 (Lota, 1999). "99" follows the names of the cultivars analyzed in 1999 while those analyzed in 2016 are followed by "16".

## 4. Discussion

### 4.1. Organization of the Genetic Diversity of the Citrus Family Group

The genetic diversity of the selected citrus cultivars was found to be closely related to the 4 ancestral species: pummelo, mandarin, citron and *C. micrantha*. The genetic diversity structure was consistent with hypotheses on *Citrus* phylogeny [11,42,43,53,55]. Secondary species resulting from interspecific hybridization, including sour orange, orange, lemon and lime, were positioned between their spawning species, i.e., grapefruit, sour orange, with orange positioned between mandarins and pummelos, while limes were close to citron and *C. micrantha*. Clementine was at equal distance from "Willowleaf" mandarin and sweet orange—its two genitors—and lemon was between citron and sour orange (Figure 2). "Nasnaran" is not a pure mandarin but rather a hybrid between mandarin and *C. micrantha* [68], which was in line with its position in the factorial analysis (Figure 2). Genetic distances between mandarin cultivars were greater than those between the other ancestral taxa according to a complex evolution process based on intra- and interspecific hybridizations that resulted in pummelo genome introgression [69]. The off-center position of Page and Fairchild relative to the true mandarin cluster ("Dancy", "Sunki", "Willowleaf") could be explained by their relationship with two tangelos (mandarin x grapefruit) that are pollinators of clementine mother trees [46].

The diversity analysis clearly showed the broad genetic diversity of the citrus family group as well as the one of mandarin family group. It could thus be assumed that the genetic part of the variation of the EO composition is important and may partially mask the effect of environmental variation. Therefore, in a second step, the study was carried out on a reduced sampling, with only two groups (mandarin and citron families) and by removing "Nasnaran" from the mandarin group in order to limit the effect of genetic variation and showcase the effect of environmental variation.

### 4.2. Environmental Effect on EO Yield and Composition

In Corsica, the chemical profiles of EOs observed in the current study were closely identical to those obtained 20 years earlier for the majority of the cultivars from the INRAE-CIRAD citrus germplasm collection [66]. The only environmental factor that changed between the present study and those carried out between 1996 and 1998 was temperature, with an increase of 0.4 to 1°C over the entire fruit development period. This climate change over the past 20 years has likely altered some of the physicochemical characteristics of clementine, such as the earlier decrease in acidity and the delay in skin coloration [70]. However, some variations in chemical profiles were observed in the 20-year interval between the two studies on PEOs from "Etrog" citrus and LEOs from "Sunki" mandarin. Unfortunately, no traceability records were available to identify the trees and cultivars from the 1999 study [66]. "Etrog" is not a citrus cultivar but rather the name of a group of citron cultivars whose fruit are used in the Feast of Tabernacles [18]. Similarly, there are several "Sunki" mandarin accessions in the INRAE-CIRAD citrus collection with great morphological and genetic differences between them [44]. It could therefore be assumed that the variation in EO composition of these cultivars ("Etrog" and "Sunki") are related to a genetic dissimilarity. For the other cultivars, the increasing temperature in Corsica did not seem to affect the overall EO composition.

Differences in EO yields were mainly observed on the fruit skin, with a slightly higher average yield noted for cultivars grown in Corsica compared to the same cultivars cultivated in Bahia. On a case-by-case basis, the opposite was true for few cultivars. However, these differences could be explained by the fact that in the case of mandarins, i.e., easy peeler fruit, the entire fruit peel was collected while for the other citrus cultivars such as oranges, lemons, citron and grapefruit, the sampling was done with a knife by cutting the external part of the fruit peel (flavedo). For these latter citrus knife-peeled samples, depending on the experimenter, the thickness of the peelings may vary with various extents of albedo collected, thus impacting the EO yield. This bias cannot would not apply for mandarins and no differences were observed between the fruit skin thickness of mandarins related to

their geographical origin [45]. This suggests that the differences observed in mandarin PEO yields were only related to the environmental effects.

For manually peeled fruit, the "Nasnaran" cultivar had the highest PEO yield. In addition, "Nasnaran" fruit were the smallest fruits of all the cultivars studied here (about 2 cm in diameter for a mass of about 5–10 g) so the resulting fruit peel yield per kilogram of fruit was the highest of all.

Volatile organic compounds (VOCs) are often considered to be a response mechanism for plants to stresses in order to cancel or decrease their negative consequences [71]. VOC profiling has been used to evaluate biotic and abiotic stress responses in citrus plants to: Citrus Tristeza Virus [72], HLB disease [35,37], winter flooding and salinity [73], blue light [74], chilling temperatures [75] or water deficits [76]. The hypothesis of a disease factor that would explain the observed differences in EO yield and composition would likely not apply in our study since the selected trees were in good health without any physiological disorder symptoms.

Three major factors remain, i.e., cultivation practices, soil and climate. "Rangpur" lime is widely used in Brazil as a rootstock related to its ability to provide drought tolerance to the grafted cultivar, mainly orange tree. The rainy season corresponds to the fruit development period when tree water requirements are the highest. In the Mediterranean zone, the citrus fruit development period corresponding to phases I and II occurs during a dry season, and the drought tolerance of "Rangpur" lime would not enable good fruit development. Water requirements are then fulfilled by irrigation in this area. The effect of the rootstock on EO yield and composition has already been described in mandarin/orange/grapefruit [34], bergamot [31], "Page" mandarin [33] and kumquat [32]. However, it is quite variable and sometimes absent in certain combinations and even very marked in others. Previous studies on the citrus fruit in the INRAE-CIRAD collection of San Giuliano revealed that the orange EO composition was stable regardless of the rootstock, i.e., Carrizo citrange, trifoliate orange or Volkamer lemon [14–16]. These results indicate that the effect of rootstocks on the flavor of citrus fruit is a rather complex phenomenon that greatly depends on specific interactions between the rootstock, the scion cultivar and the environment.

Fiuza et al. [77] studied the relationship between variations in the chemical composition of citron EOs and the macro- and micronutrient (iron, manganese, calcium, etc.) composition of the soil and finally concluded that there was no statistic correlation between these two variables. Nevertheless, numerous studies in several aromatic plants revealed that the nature of the soil, i.e., its structure and pH, can modify the EO composition. The pH is an EO variation factor in *Rosmarinus officinalis* L. [78]. Soil acidity increases the proportion of myrcene and decreases the proportion of β-curcumene in the EO of *Eryngium campestre* L. [79]. Elevation is a factor that may also be responsible for variations, especially in the concentration of some monoterpenes in the EO of *Helichrysum italicum* ssp., whereas the soil pH was reported to have no effect on the EO of this plant [80,81]. The EO of *Thymus spinolosus* Ten. was found to have higher amounts of monoterpenes when these plants were growing in calcareous soil as compared to those growing in siliceous soil [82].

The climate at both sites was very different with regard to both temperature and precipitation levels. Higher average temperatures and radiation are known to decrease EO yields. For example, studies of *O. basilicum* EOs suggested that high temperatures decrease oxygenated monoterpene and sesquiterpene contents, as well as EO yields [83,84]. In agreement with these results, under Corsican conditions we measured higher EO yields, proportions of oxygenated sesquiterpenes in LEOs and monoterpene contents in PEOs of acidic citrus than under Bahian conditions. Limited variability was also observed between the chemical profiles of *C. maxima* EOs from five different geographical locations in south India [85], but no information was given on the different environmental conditions or on the cultivar identity and similarity.

A water deficit was previously found to reduce the total EO production in *S. officinalis* [86], parsley [87], and Mexican oregano [88] while it improved the EO yield in sage [89]. In *Pituranthos scoparius* EO from three regions in Algeria, there is likely a closer relationship between the oil composition and the annual rainfall than between this composition and the temperature levels [90].

Even though the water regime differed between San Giuliano (Corsica) and Cruz das Almas (Bahia), it cannot be concluded that the trees of the Cruz das Almas citrus collection were in a water deficit situation at the time of the fruit and leaf harvests.

*4.3. Consequences of Changes in EO Composition*

Although the observed variations in aromatic composition may have an impact on the fragrance and aroma of EOs and derivatives, the cosmetic or perfumery industry may benefit from the results of this study to select the raw material origin, i.e., EO. For instance, the difference in the proportion of methyl N-methylanthranilate noted between Corsica (1.10%) and Bahia (0.35%) in PEOs of Willowleaf mandarin could strengthen or weaken the typical aroma of the Mediterranean mandarin *C. deliciosa* [91,92] depending the EO origin. On the other hand, limonene enhances the "fruity zesty" character and the fresh flavor. This character would therefore be more pronounced in EO produced in Brazil than in the Mediterranean zone. Nootkatone—a compound that is highly sought by the cosmetic industry for its typical grapefruit odor—was only detected in Corsican grapefruit PEO. The β-pinene that has a "pine resin" aroma, was in proportion around 5-fold higher in EOs of "Tahiti" lime and "Eureka" lemon from Corsica than in EOs from Bahia. For this same cultivar, citrus cultivation in Corsica also favors higher proportions of compounds from the family of the geranial family, thus generating "rose" and "geranium" fragrances. Otherwise, the proportions of geranial (lemon odor) and geraniol (flowery-roselike odor) in LEOs of "Feminello" and "Eureka" lemons were 2-fold higher in trees cultivated in Bahia than in Corsica.

## 5. Conclusions

The present study demonstrated the effect of the environment and the rootstock, on the composition and yield of citrus fruit leaf and fruit peel EOs for all of the studied crop cultivars. The variation factors of these two environments were too numerous (temperature, solar radiation, hygrometry, rootstocks, etc.) to identify those responsible for the observed variations. Changes in EO composition were more marked in the citron family than in mandarins. The proportions of monoterpenes esters and oxygenated sesquiterpenes were higher in Corsica than in Bahia. Sensory analyzes on EOs would be necessary to evaluate the organoleptic changes in relation to these chemical differences. This would help improve management of material supplies according to the climatic conditions and technical practices in order to obtain essential oils with different aromatic profiles corresponding to the expectations of the cosmetic industry.

**Supplementary Materials:** The following are available online at http://www.mdpi.com/2073-4395/10/9/1256/s1, Supplementary File 1. PCA of the chemical diversity of the citron family based on PEO components (left) and the contribution of each compound to the diversity of the considered cultivars (right). The geographical location is distinguished by different colors and letter codes, i.e., blue and ZC for Corsica and red and ZB for Bahia. Supplementary File 2. PCA of the chemical diversity of the mandarin family based on PEO compounds (left) and the contribution of each compound to the diversity of the considered cultivars (right). The geographical location is distinguished by different colors and letter codes, i.e., blue and ZC for Corsica and red and ZB for Bahia.

**Author Contributions:** Conceptualization, F.L. and F.M.; methodology, F.L., F.T.; formal analysis, C.G.N., F.L., G.C. and M.P.; resources, A.d.S.G.; writing original draft preparation, F.L.; writing—review and editing, F.L., F.M., P.O., F.L. and M.G.; supervision, F.L., F.M., M.G.; funding acquisition, F.M., F.L. and F.T. All authors have read and agreed to the published version of the manuscript.

**Funding:** This research was supported by the Pesquisador Visitante Especial (PVE) program of the Conselho Nacional de Desenvolvimento Científico e Tecnológico (CNPq), by Coordenação de Aperfeiçoamento Pessoal de Nível Superior (CAPES) and Agropolis Fondation (CAPES-Agropolis "Mandarin quality" project). CGN was funded by CAPES. FM and ASG received a Productivity Grant from CNPq (PQ). This work was conducted in the framework of the International Consortium in Advanced Biology (CIBA).

**Acknowledgments:** The authors thank Simone Teles and the Universidade Federal do Recôncavo da Bahia (UFRB, Cruz das Almas, Brazil) for supplying the laboratory infrastructure for EO extraction in Brazil, Tibério San tos Martins da Silva (EMBRAPA, Cruz das Almas, Brazil) for climate data. The authors also thank Pailly Olivier, Director of the INRAE Citrus Experimental Unit (San Giuliano, France) for providing access to citrus genetic resources in the INRAE-CIRAD citrus BRC and Orlando Sampaio Passos for providing access to citrus genetic resources at EMBRAPA (Cruz das Almas, Brazil).

**Conflicts of Interest:** The authors declare no conflict of interest.

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
