# Peer review of "Effect of Environmental Conditions on the Yield of Peel and Composition of Essential Oils from Citrus Cultivated in Bahia (Brazil) and Corsica (France)"

_agronomy, doi:10.3390/agronomy10091256_

Round 1

Reviewer 1 Report

The paper of François et al. investigates the effect of the cultivation area on the essential oils in a variety of citrus genotypes. The topic is interesting. The manuscript is well written and the presentation of the results requires only a slight revision.

Points for improvement:

Lines 117-120. Please clarify

The manuscript is loaded with data, for convenience please consider presenting some of the figures (ie Figures 4 and 5) as supplementary data for this manuscript.

A small comment about the effect of the region climate on the essential oil-related quality should be beneficial for the discussion part of this manuscript.

Author Response

We would like to thank the proofreaders for their valuable work in improving our manuscript. Our answers are written in blue.

Lines 117-120. Please clarify

 The sentence was amended to clarify the idea that the addition of the Mexican lime in the study of genetic diversity was justified by the fact that it is a relative of the Tahiti lime.

The manuscript is loaded with data, for convenience please consider presenting some of the figures (ie Figures 4 and 5) as supplementary data for this manuscript.

 We agree with the idea to put some figures as additional data;  we have selected figures 7 and 8 to be put in additional data because they do not show very clear differences in the effect of the environment on the composition of fruit essential oils.

A small comment about the effect of the region climate on the essential oil-related quality should be beneficial for the discussion part of this manuscript.

As suggested by reviewers 1 and 3, we have expanded the discussion on the effect of the environment on essential oil composition with the following additional information:

Fiuza et al. [77] studied the relationship between variations in the chemical composition of citron EOs and the macro- and micronutrient (iron, manganese, calcium, etc.) composition of the soil and finally concluded that there was no statistic correlation between these two variables. Nevertheless, numerous studies in several aromatic plants revealed that the nature of the soil, i.e. its structure and pH, can modify the EO composition. The pH is an EO variation factor in Rosmarinus officinalis L. [78]. Soil acidity increases the proportion of myrcene and decreases the proportion of b-curcumene in the EO of Eryngium campestre L. [79]. Elevation is a factor that may also be responsible for variations, especially in the concentration of some monoterpenes in the EO of Helichrysum italicum ssp., whereas the soil pH was reported to have no effect on the EO of this plant [80, 81]. The EO of Thymus spinolosus Ten. was found to have higher amounts of monoterpenes when these plants were growing in calcareous soil as compared to those growing in siliceous soil [82].

Limited variability was also observed between the chemical profiles of C. maxima EOs from five different geographical locations in south India [85], but no information was given on the different environmental conditions or on the cultivar identity and similarity.

In Pituranthos scoparius EO from three regions in Algeria, there is likely a closer relationship between the oil composition and the annual rainfall than between this composition and the temperature levels. [90].

Reviewer 2 Report

Line 2 Change the title by removing word yield and replaced with “The yield of peel and composition of leaf essential oils” because yield means the fruits. The introduction needs to focus more on the effect of environmental conditions on yield and composition. Also, the authors did not mention which compositions are important. Line 96, what authors mean by water status? Line 110, Plant Material: it is so complicated to understand. It needs to be clear. Also, my suggestion is to move table 1 direct after the plant material part. Line 125 Climate, soil, and cultivation practice description, figure 1 needs to move directly after this part. Line 133, have the authors recorded the amount of water that gave to trees? Line 147, table 2 needs to move after “Genetic conformity of citrus varieties between EMBRAPA and INRAE-CIRAD germplasm collections verified by molecular marker genotyping” part line 226, spaces after t, =, p, and =. Also, p needs to be in italic style. Line 227 spaces after t, =, p, and =. Also, p needs to be in italic style. All tables and figures need to be directly after the part that mentions each one of them. Figure 1 it is not clear what are B, in yellow and blue, FH, and LH, in yellow and blue. 421, 424, and 429 tables 2, 3, and 4, the font type needs to change. Line 467 Does 0.6 C increasing in temperature make the difference in the other studies?

Author Response

We would like to thank the proofreaders for their valuable work in improving our manuscript. Our answers are written in blue.

Line 2 Change the title by removing word yield and replaced with “The yield of peel and composition of leaf essential oils” because yield means the fruits.

The correction was made.

The introduction needs to focus more on the effect of environmental conditions on yield and composition. Also, the authors did not mention which compositions are important.

We don't understand the reviewer's criticism and request. The introduction is constructed as follows: (1) a presentation of the biological model, i.e. citrus fruits, with their different growing environments and the effects in particular of temperature on the composition of primary and secondary metabolites (22 lines); (2) Then a focus on EOs and the different factors of variation in their composition, including environmental factors (42 lines); (3) finally, 9 lines present the objectives and the strategy of the present research work. The entire introduction focuses on the effect of the environment on the ripening, colouring and biochemical composition of citrus fruits. In addition, two thirds of the introduction (66%) is a bibliographical review, with 32 cited references, of the variation in the EO composition of citrus fruit according to different factors, mainly environmental factors. Environmental factors are not only climatic factors but includes also soil composition, cultivation practices (such as rootstock, fertilization, irrigation…), pathogens etc… We believe that it is necessary in an introduction to make convergence towards the main theme of study and not to limit the bibliographical review only to the subject of study. This makes it possible to place the research question in a general problematic or a more global context. On the other hand, on citrus there are few suitable studies on the subject, because often the environments and growing techniques are different and the genetic similarity of the cultivars is not verified.

We don't think we can focus even more on “the effect of environmental conditions on yield and composition”.  

The importance of an EO composition is a relative notion.  It depends on the end use of the raw material (fruit or EO) and the expectations of the users.  In the processing industry, the main objective is to obtain standardization or stability of the commercial product in terms of taste if it is a food product or in terms of aroma for a cosmetic product. This study does not aim to demonstrate that it is better to obtain raw material from a particular origin but that depending on the origin, it is possible to obtain EO with different compositions and therefore potentially with different aromatic profiles. it is rare to find in the exploitation of citrus EO an aromatic identity linked to a geographical origin, contrary to other fruits such as vanilla for example. Under these conditions, how do you want to define the importance of an EO composition?

Perhaps the question of importance concerns the plant organ, fruit or leaf? In this case, we have added a sentence in the last paragraph of the introduction explaining why the analysis of EOs was done using leaves and fruits. We added the followed sentences:

Line 60: “Although fruit EO is mostly used by industry, leaf EO, called "petit grain" is also used in cosmetics.” and

Line 70: These studies showed that leaf-extracted EO (LEO) is better suited to study diversity and taxonomy than fruit skin EO (PEO) because limonene is present in much lower proportions in LEO than in PEO, making it easier to observe variation of other compounds.

Line 96, what authors mean by water status?

It is not a water status referenced in the bibliography but rather of water level (in the soil or supplied by precipitation or irrigation). We corrected the text by changing the word.

Line 110, Plant Material: it is so complicated to understand. It needs to be clear. Also, my suggestion is to move table 1 direct after the plant material part.

The description of the plant material used for studies on genetic diversity and the composition of essential oils has been modified to make it more understandable. 

However, the position of figures and tables in the manuscript followed the recommendations for authors requested by the Agronomy journal. They should be grouped in one section at the end of the results paragraph. It is possible (and I hope) that in the final version of the manuscript, the distribution of the tables and figures will be closer to their citation in the text.

Line 125 Climate, soil, and cultivation practice description, figure 1 needs to move directly after this part.

We agree with you, but we followed the recommendations of the review.

Line 133, have the authors recorded the amount of water that gave to trees?

We haven't recorded the amount of water given to the trees.

Line 147, table 2 needs to move after “Genetic conformity of citrus varieties between EMBRAPA and INRAE-CIRAD germplasm collections verified by molecular marker genotyping” part

We agree with you, but we followed the recommendations of the review for authors.

line 226, spaces after t, =, p, and =. Also, p needs to be in italic style.

These corrections were made.

Line 227 spaces after t, =, p, and =. Also, p needs to be in italic style.

These corrections were made.

All tables and figures need to be directly after the part that mentions each one of them.

We agree with you, but we followed the recommendations for authors.

Figure 1 it is not clear what are B, in yellow and blue, FH, and LH, in yellow and blue.

We had forgotten to specify what the shortcuts represent (the letters and colors in the legend of figure 1). We modified the legend as follows:

Blooming (B) as well as fruit and leaf harvesting (FH and LH) periods at each station are indicated at the top of the graphs, in yellow for Cruz das Almas and in blue for San Giuliano

421, 424, and 429 tables 2, 3, and 4, the font type needs to change.

We changed the font type.

Line 467 Does 0.6 C increasing in temperature make the difference in the other studies?

We are not aware of any other studies on the effects of climate change on citrus fruit quality.  A fellow researcher at the INRAE station in San Giuliano has been studying the relationship between climate and clementine colouring/acidity over the last 30 years.  An increase of about 1°C during phase I of fruit development, compared to the average temperature recorded in the 1990s, delays skin coloration by about 3 weeks and accelerates the decrease in juice acidity (unpublished). Thanks to these observations, he has developed a computer model for predicting the date of ripening and helping to schedule the clementine harvest in Corsica, which is accessible to farmers.  

Reviewer 3 Report

The paper discusses the differences in the growing and chemical composition of citrus fruit in different climatic regions. The work is interesting and I congratulate the authors the idea. The work requires minor corrections, details of which are given below.

Abstract:

Line 23: The noun “fruit” has on only the singular form. Please check it in the whole paper.

Line 24-25: Names of markers used for the first time should be developed

Line 24 and 29 and many others below: The paper is not about wild plants: the designation “variety” should be replaced by “cultivar” (or, as the case may be, “species” or “genus”)

Introduction:

Line 38: you should avoid the expression “negative temperatures” because it has different meaning depending of the scale. Freezing temperatures? Temperatures below zero? Below-zero temperatures?

Line 65 and 68: see comment #3 in the abstract

Line 76: Please use the full scientific name in brackets when the name is used for the first time (like you did before)

Line 77-78: as above (C macrophylla)

Line 80: sci name of hybrid

Line 80: Troyer as a cultivar should be in single quotation marks (see International Code of Nomenclature for algae, fungi, and plants)

Line 82: “Page mandarin” as above

Line 87: Candidatus Liberibacter

Material and Methods

Line 111: cultivars

Line 113: Please be more precise. I guess 8 species were supplemented by three others or 17 cultivars were supplemented by 7 cultivars (specimens or accessions).

Line 146: It is desirable to add some information about canopy management procedures, which might be different in different countries.

Line 175 & 190: All equipment like Clevenger-type apparatus or Perkin-Elmer Autosystem GC apparatus and others should have details about the manufacturer, country etc.

Results:

Line 217: cultivars

Line 225: You cannot state that something was higher if you did not carry out a variance analysis. There are certain compounds for which you obtained the opposite.

Line 370: the shortcuts presented in the figure should be developed below it.

Conclusions:After reading the discussions and conclusions, I feel an insufficient discussion of the influence of the climatic and environmental conditions on the chemical composition of fruit and a poor discussion with the literature on this subject. If there is no such work on citrus fruit, then perhaps the literature contains studies on other fruit species cultivated in different climatic conditions and the influence of those conditions on the chemical composition of fruit.

Author Response

We would like to thank the reviewers for their valuable work in improving our manuscript. Our answers are written in blue.

Abstract:

Line 23: The noun “fruit” has on only the singular form. Please check it in the whole paper.

the correction was made

Line 24-25: Names of markers used for the first time should be developed

Made

Line 24 and 29 and many others below: The paper is not about wild plants: the designation “variety” should be replaced by “cultivar” (or, as the case may be, “species” or “genus”)

The correction has been made in all the ms. However, how do you consider a wild citrus fruit whose origin is, for example, a natural hybridization, and which is cultivated? Cultivar or variety? This is the case of many citrus fruits such as Tahitian lime or Mexican lime, which have not undergone any human selection.

Introduction:

Line 38: you should avoid the expression “negative temperatures” because it has different meaning depending of the scale. Freezing temperatures? Temperatures below zero? Below-zero temperatures?

We've changed “negative temperatures” by “freezing temperatures”

Line 65 and 68: see comment #3 in the abstract

The correction has been made in all the ms

Line 76: Please use the full scientific name in brackets when the name is used for the first time (like you did before)

The correction was made in all the manuscript

Line 77-78: as above (C macrophylla)

We added the scientific name

Line 80: sci name of hybrid

We added the scientific name

Line 80: Troyer as a cultivar should be in single quotation marks (see International Code of Nomenclature for algae, fungi, and plants)

We have added single quotation marks to all cultivar names in the manuscript

Line 82: “Page mandarin” as above

We have added single quotation marks to all cultivar names in the manuscript

Line 87: Candidatus Liberibacter

 We modified

Material and Methods

Line 111: cultivars

We modified

Line 113: Please be more precise. I guess 8 species were supplemented by three others or 17 cultivars were supplemented by 7 cultivars (specimens or accessions).

We modified the sentence as followed: “The batch of 17 cultivars was supplemented by 5 additional cultivars representing 3 Citrus species…”.

Line 146: It is desirable to add some information about canopy management procedures, which might be different in different countries.

We understand by "canopy management procedures", tree-pruning techniques. If this is the case, we do not think that the pruning technique is very different in our 2 geographical sites, because the management of the canopy of the trees in a collection is essentially carried out to facilitate the passage of agricultural machinery, i.e. by cutting down the low branches. The situation is very different in a production orchard where the pruning techniques are adapted to the cultivars (for example a clementine tree is not pruned like a lemon tree) and adapted to the harvesting process which depends on the use of the fruit (fresh fruit market or juice industry). Since our study based on biological material taken from collection trees, we did not consider that the pruning technique was a factor in the variation of the composition of the essential oil.

Line 175 & 190: All equipment like Clevenger-type apparatus or Perkin-Elmer Autosystem GC apparatus and others should have details about the manufacturer, country etc.

 This information has been added

Results:

Line 217: cultivars

Modified

Line 225: You cannot state that something was higher if you did not carry out a variance analysis. There are certain compounds for which you obtained the opposite.

In this paragraph of the “results“ section we have analyzed the quantity of EOs extracted from leaves and fruits in relation to the same unit of mass of treated plant tissue; it is therefore a yield. We did not compare the yield of the compounds. On the other hand, we have done a statistical treatment, which proves that differences at the level of a type of citrus family were significant (t-test). We do not excluded that the result could be the opposite at the level of the cultivar. 

“It should be noted that ‘Tahiti’ lime and ‘Fairchild’ mandarin produced a 2-fold higher quantity of EOs in the peel of fruit produced in Bahia than in Corsica”

Line 370: the shortcuts presented in the figure should be developed below it.

We had forgotten to specify what the shortcuts represent (the letters and colors in the legend of figure 1). We modified the legend as follows:

Blooming (B) as well as fruit and leaf harvesting (FH and LH) periods at each station are indicated at the top of the graphs, in yellow for Cruz das Almas and in blue for San Giuliano

Conclusions:After reading the discussions and conclusions, I feel an insufficient discussion of the influence of the climatic and environmental conditions on the chemical composition of fruit and a poor discussion with the literature on this subject. If there is no such work on citrus fruit, then perhaps the literature contains studies on other fruit species cultivated in different climatic conditions and the influence of those conditions on the chemical composition of fruit.

As suggested by reviewers 1 and 3, we have expanded the discussion on the effect of the environment on essential oil composition. We believe that in the discussion we should remain on the subject of the effect of the environment on the EO composition and not on the quality of the fruit, which is a separate subject, which we mentioned a little bit in the introduction and in the discussion. We added the following sentences to the discussion section:

Fiuza et al. [77] studied the relationship between variations in the chemical composition of citron EOs and the macro- and micronutrient (iron, manganese, calcium, etc.) composition of the soil and finally concluded that there was no statistic correlation between these two variables. Nevertheless, numerous studies in several aromatic plants revealed that the nature of the soil, i.e. its structure and pH, can modify the EO composition. The pH is an EO variation factor in Rosmarinus officinalis L. [78]. Soil acidity increases the proportion of myrcene and decreases the proportion of b-curcumene in the EO of Eryngium campestre L. [79]. Elevation is a factor that may also be responsible for variations, especially in the concentration of some monoterpenes in the EO of Helichrysum italicum ssp., whereas the soil pH was reported to have no effect on the EO of this plant [80, 81]. The EO of Thymus spinolosus Ten. was found to have higher amounts of monoterpenes when these plants were growing in calcareous soil as compared to those growing in siliceous soil [82].

Limited variability was also observed between the chemical profiles of C. maxima EOs from five different geographical locations in south India [85], but no information was given on the different environmental conditions or on the cultivar identity and similarity.

In Pituranthos scoparius EO from three regions in Algeria, there is likely a closer relationship between the oil composition and the annual rainfall than between this composition and the temperature levels. [90].